# The impact of sea spray aerosol on photochemical ozone formation over eastern China: heterogeneous reaction of chlorine particles and radiative effect

Yingying Hong[1,3], Yuqi Zhu[1,2], Yuxuan Huang[1,2], Yiming Liu[1,2,*], Chuqi Xiong[1,2], Qi Fan[1,2,*]

[1]School of Atmospheric Sciences, Sun Yat-sen University, and Key Laboratory of Tropical Atmosphere-Ocean System, Ministry of Education, Zhuhai, China

[2]Guangdong Provincial Observation and Research Station for Climate Environment and Air Quality Change in the Pearl River Estuary, Southern Marine Science and Engineering Guangdong Laboratory (Zhuhai), Zhuhai, China

[3]Guangdong Ecological Meteorology Center, Guangzhou, China

*Correspondence to*: Yiming Liu (liuym88@mail.sysu.edu.cn) and Qi Fan (eesfq@mail.sysu.edu.cn)

**Abstract.** Eastern China has suffered from severe photochemical $O_3$ (ozone) pollution in recent years. In this coastal region, atmospheric environment can be influenced by sea spray aerosol (SSA) from marine emissions. However, the extent and mechanisms by which SSA affects $O_3$ formation remain incompletely understood. Here, using the WRF-CMAQ model, this study investigates the comprehensive effect of SSA on radical chemistry and $O_3$ formation in the lower troposphere across four seasons. SSA (over 50% are particulate chlorine) can reach further inland through an atmospheric "bridge" aloft, interacting with the nitrogen-containing gases from continental anthropogenic emissions to reduce $NO_x$ levels and release Cl radicals. The $NO_x$ reduction increases $O_3$ in VOCs-limited regions while decreasing them in $NO_x$-limited zones. Elevated Cl radicals enhance VOCs degradation and $O_3$ formation during morning hours. Meanwhile, the scattering properties of SSA reduce daytime $O_3$ formation by diminishing photolysis rates. Due to the contrasting effect of SSA via different mechanisms, the response of $O_3$ varies seasonally and geographically. In winter, SSA increases $O_3$ in eastern China due to the dominant effect of $NO_x$ reduction in VOCs-limited regions. In spring and autumn, similar effects occur in North China Plain, whereas southern China sees a decrease due to the $NO_x$ reduction in $NO_x$-limited region and reduced photolysis rates. In summer, $O_3$ increases are observed only around Bohai, with reductions elsewhere driven by $NO_x$ reductions in $NO_x$-limited regions and decreased photolysis. This study highlights the important, varying, but previously unreported role of SSA in shaping tropospheric photochemistry over eastern China.

# 1 Introduction

In recent years, eastern China has grappled with severe photochemical $O_3$ (ozone) pollution, eliciting widespread concern from governmental and academic sectors (Wang et al., 2022; Lu et al., 2018; Wang et al., 2017). Elevated $O_3$ concentrations pose serious threats to public health and have detrimental effects on vegetation growth and crop yields (Fleming et al., 2018; Lefohn et al., 2018; Liu et al., 2018a; Feng et al., 2022). Traditionally perceived as a warm-season issue (Lu et al., 2020), $O_3$ pollution has recently been documented during winter and spring, driven by substantial reductions in $NO_x$ emissions juxtaposed against relatively stable VOC (volatile organic compounds) emissions in China since 2013 (Li et al., 2021). The escalations of $O_3$ levels following the Clean Air Action in 2013 underscores the need for a nuanced understanding of its formation mechanisms throughout the year (Liu and Wang, 2020b, a; Wei et al., 2022).

Sea spray aerosol (SSA) is a major natural particulate source in the atmosphere (Weis and Ewing, 1999; Roth and Okada, 1998), generated from oceanic surfaces through wave-breaking and bubble-bursting processes (Lewis and Schwartz, 2004). According to IPCC reports, an estimated 3300 Tg of SSA annually enters the atmospheric boundary layer, driven by wind stress on ocean surfaces, contributing to complex atmospheric chemical interactions. The rapid economic growth in coastal areas, coupled with unique challenges of complex atmospheric pollution, particularly in eastern China, has highlighted the role of SSA in atmospheric oxidation processes. While previous studies have indicated the potential impact of SSA on coastal $O_3$ levels (Dai et al., 2020; Knipping and Dabdub, 2003), a comprehensive and systematic investigation into the mechanisms of these effects is lacking.

SSA influences near-ground $O_3$ formation through multiple mechanisms. SSA can significantly scatter incoming solar radiation that reaches the ground, thereby diminishing the actinic flux within the troposphere (Lohmann and Feichter, 2005; Hatzianastassiou et al., 2007). This reduction in solar energy lowers the rate of photochemical reactions crucial for $O_3$ generation. Near-ground $O_3$ primarily forms through photochemical reactions between $NO_x$ and VOCs under sunlight, with its production heavily dependent on the amount of solar radiation penetrating to the surface. As a prominent source of atmospheric aerosols, SSA can attenuate this solar radiation, leading to decreased photolysis rates of $NO_2$ and, consequently, reduced $O_3$ formation (Li et al., 2011; Xing et al., 2017). Currently, this mechanism—the impact of SSA on ground-level ozone through the modulation of photolysis rates—is often overlooked in previous studies on the influence of SSA on $O_3$ concentrations.

Besides, particulate chlorine in SSA engages in heterogeneous chemical reactions with nitrogen-containing gases, releasing Cl radicals that enhance atmospheric oxidation and affect photochemical $O_3$ formation. While the traditional pathway for urban $O_3$ formation involves the reaction of VOCs with hydroxyl radicals (OH), recent research underscores the significant role of

Cl radicals in similar processes (Faxon and Allen, 2013; Qiu et al., 2019; Young et al., 2014). Chlorine radicals react with VOCs more rapidly than OH radicals, despite their lower atmospheric concentrations, making their oxidation potential comparably significant (Aschmann and Atkinson, 1995; Nelson et al., 1990; Wingenter et al., 1999).

Particulate chlorine reacts with $N_2O_5$ to form nitroxyl chloride ($ClNO_2$), which releases chlorine radicals upon photolysis, contributing significantly to atmospheric Cl levels (Thornton et al., 2010; Bertram and Thornton, 2009; Roberts et al., 2009). These reactions are outlined in R1 and R2 (where g and cd represent the gas and condensed phases, respectively). Observations of $ClNO_2$ have shown high concentrations in eastern China, indicating active chlorine chemistry (Tham et al., 2016; Yun et al.,

2018; Wang et al., 2016).

$$N_2O_5(g) + Cl^-(cd) \rightarrow ClNO_2(g) + NO_3^-(cd) \tag{R1}$$
$$ClNO_2(g) + hv \rightarrow Cl \cdot (g) + NO_2(g) \tag{R2}$$

Particulate chlorine can also react with $NO_2$ to produce nitrosyl chloride ($ClNO$), which, like $ClNO_2$, can release Cl radicals (R3 and R4) under photolysis (Finlayson-Pitts, 2003; Faxon and Allen, 2013).

$$2NO_2(g) + Cl^-(cd) \rightarrow ClNO(g) + NO_3^-(cd) \tag{R3}$$
$$ClNO(g) + hv \rightarrow Cl \cdot (g) + NO(g) \tag{R4}$$

Furthermore, particulate chlorine can also directly react with $NO_3$ to release Cl radicals (R5), which is a potentially important nighttime source of Cl radicals (Gershenzon et al., 1999; Seisel et al., 1999).

$$NO_3(g) + Cl^-(cd) \rightarrow Cl \cdot (g) + NO_3^-(cd) \tag{R5}$$

Most chemical transport models only considered the reaction between particulate chlorine and $N_2O_5$, neglecting other significant chlorine-related heterogeneous chemical processes. To accurately assess the effect of SSA on $O_3$ concentration, it is essential to integrate recent findings on the heterogeneous chemistry of chlorine and update chemical models accordingly. This integration will provide a more comprehensive understanding of SSA's role in photochemical $O_3$ generation, addressing

a critical gap in current atmospheric chemistry research.

Heterogeneous chemical reactions of SSA with $NO_x$ can alter $NO_x$ concentrations, thereby affecting $O_3$ production. As illustrated by reactions R1-R5, nitrogen oxides undergo heterogeneous chemical reactions with particulate chlorine at night,

leading to reduced $NO_x$ concentrations. By morning, the photolysis of ClNO and $ClNO_2$ results in the release and subsequent increase $NO_x$ concentrations. The relationship between $O_3$ and its precursors is highly nonlinear and varies by region and time due to the differences in the $O_3$ formation regime. In VOC-limited urban and suburban areas, decreases in $NO_x$ levels can paradoxically lead to an increase in $O_3$ concentration. Conversely, in $NO_x$-limited rural areas, a reduction in $NO_x$ levels typically leads to a decrease in $O_3$ production, highlighting the complex dynamics of atmospheric chemistry (Wang et al., 2017).

In terms of model-based studies, Knipping and Dabdub (2003) incorporated SSA emissions into their model, finding that $O_3$ concentration in the coastal areas of California in the United States increased by 12 ppb in the morning and 4 ppb at noon. Similarly, Sarwar and Bhave (2007) utilized the Community Multiscale Air Quality Modeling System (CMAQ) model to explore the impact of SSA emissions on $O_3$ across the eastern United States, revealing that the associated chlorine chemical processes increased the oxidation of VOCs, thereby enhancing $O_3$ production. This resulted in increases of up to 12 ppb and 6 ppb in the maximum hourly $O_3$ concentrations in the Houston and New York-New Jersey regions, respectively, and daily maximum 8-hour average $O_3$ concentrations rose by 8 ppb and 4 ppb. Dai et al. (2020) investigated the impact of $ClNO_2$ from sea-salt chloride on $O_3$ in the Pearl River Delta (PRD), China, and found an increase of up to 2.0 ppb over the inland areas during marine winds and up to 3.8 ppb and 6.5 ppb over the South China Sea. However, these studies did not fully account for the tripartite influence of SSA on $O_3$ concentration, nor did they integrate the complete heterogeneous chemistry of chlorine particulates, leading to potential uncertainties in assessing the impact of SSA.

In this study, we employed the WRF-CMAQ model to evaluate the impact of SSA on the tropospheric chemistry in eastern China during different seasons. The responses of $HO_2$, OH radicals, and $O_3$ caused by SSA were quantified. Section 2 demonstrates the modeling settings, SSA emission calculation, and experiment designs. Section 3 discusses the impact of SSA on the tropospheric chemistry. We conclude with a summary of our findings and discussions in Section 4.

**2 Methodology**

**2.1 Model settings**

Here we used the WRF-CMAQ model to perform air quality simulations in this study. The CMAQ (version 5.1) model is a regional chemical transport model developed by the United States Environmental Protection Agency (Appel et al., 2017). It has been widely used to explore the mechanism of multiple air quality issues, including tropospheric ozone, fine particles, acid deposition, and visibility degradation (Zhu et al., 2024; Kitagawa et al., 2021; Onwukwe and Jackson, 2021). The

meteorological inputs of the CMAQ model (version 5.1) were provided by the Weather Research and Forecasting (WRF) model. The configuration and emission inputs of WRF-CMAQ were consistent with those used by Hong et al. (2020). Specifically, the simulation environment was structured into two nested domains within the WRF and CMAQ models, featuring horizontal resolutions of 81 km and 27km, respectively (Fig. S1). These domains included 23 vertical layers extending up to 50 hPa. The inner domain focused on eastern China, where the detailed analysis was conducted, while the outer domain encompassed a broader area, including the land regions of East Asia and the Western Pacific. This broader scope allowed for a comprehensive consideration of SSA emission transport into the region of interest. Meteorological initial and boundary conditions for the WRF model were derived from the NCEP/NCAR final (FNL) reanalysis gridded data, which have a horizontal resolution of $1° \times 1°$. Chemical boundary conditions for the CMAQ model were sourced from the Model for Ozone and Related Chemical Tracers, version 4 (MOZART-4) results.

The calculation of photolysis rates in CMAQ uses an in-line approach for calculating actinic fluxes by solving a two-stream approximation of the radiative transfer equation (Binkowski et al., 2007; Toon et al., 1989) over wavebands based on the FAST-J photolysis model (Wild et al., 2000). Each layer includes scattering and extinction using simulated air density, cloud condensates, aerosols, and trace gaseous such as $O_3$ and $NO_2$ (Appel et al., 2017). This approach has been verified or evaluated in some previous studies. Based on the aircraft measurement, Baker et al. (2018) found that the CMAQ model can well capture the observed $NO_2$ photolysis rates at ~2km height. Using this approach, Fu et al. (2014) concluded that the $NO_2$ and $O_3$ photolysis rates reduced by up to 2.4% and 1.9% respectively, due to the impact of dust aerosol during a heavy dust event. Fan and Li (2022) also found that the $O_3$ photolysis rates decreased by 1-4% due to the extinction effect of SSA. These references provide robustness of the CMAQ model to calculate the photolysis rates. It enables us to assess the effect of aerosols (e.g., SSA) on photochemical processes by adjusting photolysis rates accordingly.

In the CMAQ simulation, we utilized the SAPRC07TIC (Carter, 2010; Hutzell et al., 2012; Xie et al., 2013) and AERO6i (Lin et al., 2013; Pye et al., 2015) mechanisms to represent gas-phase chemical and aerosol processes, respectively. The AERO6i aerosol module employed ISORROPIA (Binkowski and Roselle, 2003; Fountoukis and Nenes, 2007; Kelly et al., 2010) to uniformly simulate inorganic aerosol thermodynamics. The chlorine depletion of SSA through its equilibrium reactions with $H_2SO_4$ and $HNO_3$ was considered in the model (Liu et al., 2015). As for the heterogeneous reaction, the original model was configured to only account for the heterogeneous reaction of particulate chlorine with $N_2O_5$ (R1) (Sarwar et al., 2012; Sarwar et al., 2014). To provide a more comprehensive evaluation of SSA's impact on photochemistry, we expanded the model's capability to include heterogeneous reactions of particulate chlorine with $NO_2$ (R3) and $NO_3$ (R5). We developed a linear segmentation function to parametrize the uptake coefficients of $NO_2$ and $NO_3$ on aerosol surfaces, reflecting their strong positive correlation with relative humidity (Dentener et al., 1996; Stutz et al., 2004). This parameterization was identical across

different aerosol modes (Aitken, accumulation, and coarse). The selection of maximum and minimum uptake coefficients for $NO_2$ and $NO_3$ was based on laboratory findings, aligning with methodologies from previous modeling studies by Wang et al. (2012) and Zheng et al. (2015). A detailed exposition of these modifications and their implications for the model's chemistry of heterogeneous reactions is documented by Hong et al. (2020).

Both anthropogenic and natural emissions were incorporated in the simulation to ensure comprehensive atmospheric modeling. Anthropogenic sources included routine pollutant emissions from the MIX emission inventory (http://www.meicmodel.org/) (Li et al., 2017), international shipping emissions from HTAP (Hemispheric Transport Atmospheric Pollution) emissions version 2.0 dataset (Janssens-Maenhout et al., 2015), and chlorine emissions (HCl and $Cl_2$) from the 2014 ACEIC (Anthropogenic Chlorine Emission Inventory for China) (Hong et al., 2020; Liu et al., 2018b). For natural sources, biogenic emissions were calculated using the Model of Emissions of Gas and Aerosols from Nature (MEGAN) (Guenther et al., 2012), which utilized meteorological inputs from the WRF simulation. SSA emissions were calculated inline in the CMAQ model (see Section 2.2).

We utilized the process analysis module within the CMAQ model to quantify the production and loss of model species. This diagnostic tool employs both integrated process rate (IPR) and integrated reaction rate (IRR) methods. The IPR method assesses the contributions of various physical and chemical processes to pollutant concentrations, while the IRR method determines the contributions from specific chemical reactions. Process analysis has been extensively used in prior research to elucidate the chemical mechanism underlying $O_3$ formation (Wang et al., 2015; Chen et al., 2018; Liu et al., 2021).

**2.2 SSA emission calculation**

The SSA emission was calculated on line in the CMAQ model, utilizing a source function developed by Gantt et al. (2015), which builds upon the foundational source function established by Gong (2003). Gantt et al. (2015) introduced two main modifications to enhance the model's accuracy. Firstly, a sea surface temperature (SST) correction function was incorporated, based on the findings of Ovadnevaite et al. (2014), to account for the substantial impact of SST on SSA fluxes. This correction function linearly adjusts to SST variations, reflecting its influence on emission rates (e.g., Barthel et al., 2019; Liu et al., 2021). Secondly, the shape factor of the source function was adjusted to increase the emission of submicron SSA particles, altering the flux distribution to better reflect observed atmospheric conditions (see Fig. S1 in Gantt et al. (2015)). Additionally, surf-enhanced emissions were reduced by narrowing the defined surf zone from 50 m to 25 m from the coast, aligning with the modifications in Gantt et al. (2015). The estimated diameters of SSA range from ~0.02 μm to 20 μm in the model. The composition of dry SSA in different aerosol modes remains consistent with that of seawater, containing $Cl^-$ (55.4%), $Na^+$

(30.8%), $SO_4^{2-}$ (7.7%), $Mg^{2+}$ (3.8%), $Ca^{2+}$ (1.2%), and $K^+$ (1.1%).

## 2.3 Experiment setting

We conducted simulations for January, April, July, and October of 2015 to represent the typical atmospheric conditions for each season in eastern China. A 10-day spin-up period preceded the actual modeling to stabilize initial conditions. The simulations incorporating all emissions were designated as the baseline scenario (BASE). Additionally, a sensitivity experiment (NOSA), which excluded SSA emissions, was performed to discern the specific contributions of SSA to surface $O_3$ levels. The seasonal impacts of SSA emissions were assessed by comparing the differences in simulated pollutant

concentrations between the BASE and NOSA experiments.

Model validation was rigorously carried out against observational data from eastern China for the corresponding periods (Hong et al., 2020). The validation confirmed that the WRF-CMAQ model capably simulated key meteorological factors (temperature, relative humidity, and wind speed) and routine pollutant concentrations ($O_3$, $NO_2$, CO, $PM_{2.5}$, $PM_{10}$, and $SO_2$), along with

195 particulate chlorine concentrations. This validation provides a solid foundation for our confidence in further exploring the impact of SSA on tropospheric photochemistry using these modeling results. Detailed validation results were described by Hong et al. (2020).

## 3 Result and discussion

### 3.1 SSA transport

We used particulate $Na^+$ as a proxy for SSA, due to its major presence and relatively inactive in the atmosphere (Neumann et al., 2016). Our analysis distinguished regions significantly affected by SSA by comparing $Na^+$ concentrations from BASE and NOSA experiments (Fig. 1). Near the ocean, areas exhibiting elevated $Na^+$ align with high SSA emissions zones (Fig. S2). In eastern China, increased $Na^+$ concentrations are notably due to SSA transport from the ocean, diminishing progressively with

205 distance from the coast. Cities such as Shanghai and Guangzhou, located along the coastline of eastern China, displayed significant $Na^+$ increases (>1 μg/m³), indicating marked influence of SSA. Smaller yet notable increases (>0.1 μg/m³) extend into broader inland areas spanning several provinces including Liaoning, Tianjin, and down to Guangxi. More than 80% of the inland region's $Na^+$ within ~100 km of the coast (e.g., Shanghai, Zhejiang, Fujian, Guangdong, and Guangxi) is attributable to SSA, decreasing to 10%-60% towards central regions like Hubei. The terrain influences regional SSA impacts distinctly. In

northern China, the Taihang Mountains impede westward SSA transport, confining its influence to the North China Plain.

Conversely, lower terrain elevations in southern China facilitate broader inland SSA dispersal. Seasonally, the inland reach of SSA is most extensive in July, propelled further by the southerly summer monsoon.

SSA transport also occurs via an "aloft bridge" over the planetary boundary layer, enhancing its penetration inland. Figures 1c, 1d, and 1e illustrate vertical-diurnal variations of SSA-induced $Na^+$ concentration in Beijing, Shanghai, and Guangzhou, pinpointed in Fig. 1a. These profiles reveal that while coastal cities like Shanghai show higher surface-level concentrations, more inland cities such as Guangzhou and Beijing exhibit elevated concentrations aloft, especially in October. This pattern suggests that SSA is not only transported horizontally near the surface but also vertically mixed upward from the coast before descending inland, influenced by varying thermodynamic properties and boundary layer structures between continental and marine areas. Similar transport features were observed in northwestern Europe, where SSA influences extend approximately 400 km inland (Chen et al., 2016).

Additionally, the simulated changes in particulate $Cl^-$ concentrations due to SSA (BASE minus NOSA) are shown in Fig S3 and S4. The spatial distributions of SSA-induced $Cl^-$ near the surface mirrors that of $Na^+$ (Fig. 1). However, due to the higher composition of particulate $Cl^-$ than $Na^+$ in SSA emissions, regions experiencing >80% change in $Cl^-$ are more extensive, underlining its substantial regional impact. Notably, in cities like Beijing and Guangzhou, $Cl^-$ concentrations are higher aloft than at the surface, a distinction more pronounced than for $Na^+$. This is likely due to the higher reactivity of particulate $Cl^-$. In a polluted lower atmosphere, particulate $Cl^-$ in SSA can be chemically depleted through thermodynamic equilibrium processes and heterogeneous reactions, which will be discussed in the following section.

## 3.2 Heterogeneous reactions of particulate $Cl^-$ with nitrogen-containing gases

Particulate $Cl^-$, a chemically active and abundant component of SSA, undergoes heterogeneous reactions with $NO_2$, $NO_3$, and $N_2O_5$, releasing Cl radicals in the process. Figure 2 shows the influence of these reactions on particulate $Cl^-$ concentrations, revealing significant negative impacts along coastal regions. This suggests that SSA from the ocean mixes with nitrogen-containing gases from continental sources, leading to enhanced chlorine depletion in these areas. Seasonally, the greatest depletion occurs in January, followed by October, April, and July, likely due to variations in $NO_2$ levels. In January, a notable depletion along the eastern Chinese coastline is observed, with significant reductions in Bohai Bay, the Yangtze River Delta region, and the PRD region during April and October. In July, pronounced depletion is evident in Bohai Bay and YRD, while the PRD region shows less impact. Depletion diminishes progressively inland, with coastal areas experiencing the most significant effects.

Alongside the depletion of $Cl^-$, $NO_x$ concentration decreases due to the heterogeneous reaction between $Cl^-$ and $NO_2$, $NO_3$, and $N_2O_5$ across eastern China. Figure 3 presents the seasonal changes in $NO_x$ mixing ratios caused by SSA, showing a substantial reduction, particularly in the coastal regions of eastern China, with the most significant decreases occurring in January and the least in July. The incorporation of SSA into the model results in a decrease in $NO_x$ mixing ratios by up to 3-5 ppbv (5-10%) across different months. Given the critical role of $NO_x$ as an $O_3$ precursor, these reductions could significantly influence $O_3$ level, a topic that we will explore in detail in Section 3.5.

The reactions of particulate $Cl^-$ with $NO_2$ and $N_2O_5$ result in the production of $ClNO$ and $ClNO_2$, respectively, which are crucial precursors to Cl radicals. Figures 4a and 4b display the spatial variations in their mixing ratios induced by SSA, specifically analysed at 5:00 LST due to the nocturnal accumulation of these compounds. The figures demonstrate that SSA significantly increases $ClNO$ and $ClNO_2$ levels across eastern China, particularly in coastal regions. The spatial and seasonal patterns of these increases align closely with reductions in $Cl^-$ (Fig. 2) and $NO_x$ (Fig. 3), highlighting the important impact of SSA on these heterogeneous reactions. The most substantial effects of SSA are observed in January, followed by October, April, and July. In January, pronounced increases in $ClNO$ and $ClNO_2$ levels are noted in southeastern coastal regions of eastern China. In April and October, significant increases are localized to Bohai Bay, YRD, and PRD. In July, increases remain high in Bohai Bay and YRD. The transport of SSA inland results in diminishing increases of $ClNO$ and $ClNO_2$ from the coastline inward. Quantitatively, $ClNO$ mixing ratios increase by up to 1.0, 1.1, 1.1, 1.3 ppbv in January, April, July, and October, respectively. Over 80% of the $ClNO$ in coastal areas is sourced from SSA emissions, with some areas nearing 100% contribution (Fig. S5a). For $ClNO_2$, the maximum increases are 1.0, 0.8, 0.5, 0.8 ppbv in respective months, with a broader regional impact compared to $ClNO$ (Fig. S5b). Southern China experiences a more pronounced impact from SSA compared to the northern part, with the influence markedly tapering off towards inland regions like Hubei, Chongqing, and Guizhou where contributions decrease to around 10%.

The heterogeneous reactions between SSA and nitrogen-containing gases release Cl radicals as a result. Figure 4c shows the spatial distribution of SSA-induced Cl radical concentrations in different seasons. The increases in precursor compounds $ClNO$ and $ClNO_2$ during nighttime enhance their photolysis after sunrise, which significantly boosts Cl radical concentrations. High increases are evident in January in the Taiwan Strait, reaching up to $2.9 \times 10^4$ molecule $cm^{-3}$. The concentrations peak in April and July in the Bohai Sea, the Yellow Sea and the Taiwan Strait, with the highest increase of $4.0 \times 10^4$ and $8.1.0 \times 10^4$ molecule $cm^{-3}$, respectively. October also shows substantial increases in the Taiwan Strait and Bohai Bay. The coastal regions see Cl radical concentrations boosted by $0.2$-$2 \times 10^4$ molecule $cm^{-3}$, indicating a strong link between coastal SSA emissions and increased Cl radicals. These regions experiencing elevated Cl radical levels correspond with areas showing increases in $ClNO$ and $ClNO_2$ mixing ratios (Figs. 4a and 4b). Notably, nearly 100% of the Cl radicals in oceanic regions stem from SSA emissions,

with over 40% in eastern China attributed to the same source (Fig. S5c). The IRR process analysis module helps trace the main pathways driving this increase (Figure S6), including ClNO photolysis, ClNO$_2$ photolysis, reaction of ClO and NO, heterogeneous reaction of particulate Cl$^-$ with NO$_3$, and other processes (including Cl$_2$ photolysis, reaction of HCl and OH, and etc.). Daytime increases in Cl radicals are predominantly due to the photolysis of ClNO and ClNO$_2$ following sunrise.

Furthermore, the impact of SSA on Cl radicals is observed not only at the surface but also vertically through the atmosphere. Figure 5 examines the vertical-diurnal variations in SSA-induced Cl radical concentrations in Beijing, Shanghai, and Guangzhou. This analysis shows that SSA emissions significantly elevate Cl radical concentrations after sunrise, especially during the morning hours. These increases are more pronounced near the top of the planetary boundary layer shortly after sunrise, suggesting the impact of SSA on Cl radicals is more significant in upper levels than near the surface.

It should be noted that besides heterogenous reactions with nitrogen-containing species, particulate Cl$^-$ in SSA can react with H$_2$SO$_4$ and HNO$_3$ through thermodynamic equilibrium reactions, releasing gaseous HCl (Chi et al., 2015). HCl is another precursor of Cl radicals via its reaction with OH radicals, which generally occurs during daytime (Finlayson-Pitts, 2003). However, as shown in Fig. S6, the contribution of HCl to Cl radicals is much lower than the photolysis of ClNO and ClNO$_2$. Such small contributions of HCl were also reported in a box-model study in the North China Plain (Liu et al., 2017). It suggests the limited role of these thermodynamic equilibrium reactions in the Cl radicals and following O$_3$ formation.

**3.3 Radiative effect of SSA**

SSA plays a significant role in modulating incoming solar radiation through scattering, which influences the photolysis rates of various photochemical species. As O$_3$ formation is closely linked to the photolysis of NO$_2$ (J(NO$_2$)), examining the impact of SSA on this process is crucial. Figure 6 highlights that the J(NO$_2$) decreases by up to 15.1%, 5.7%, 6.0%, and 11.8% in January, April, July, and October, respectively, particularly in oceanic and coastal areas. This reduction in J(NO$_2$) correlates well with the spatial distributions of SSA emission (Fig. S2) and SSA-induced Na$^+$ concentration, underscoring the significant radiative effect of SSA. Despite relatively modest reductions in coastal regions (1-5%), such changes are significant enough to influence O$_3$ formation considerably. According to a study by Fan and Li (2022), similar decreases in photolysis rate (1-4%) caused by SSA led to reductions in O$_3$ mixing ratios by up to 1-2% in eastern China during July.

Additionally, the photolysis rates of photochemical gases generally increase with altitude due to the rising actinic flux (Gao et al., 2020). Figure 6 also presents the vertical-diurnal variations in J(NO$_2$) changes caused by SSA in Beijing, Shanghai, and Guangzhou. The influence of SSA on J(NO$_2$) is notably lesser in Beijing compared to the other two cities. A significant

reduction in $J(NO_2)$ is observed around noon (12:00 LST), coinciding with the daily peak in actinic flux. We note that the extinction effect of SSA can extend into the upper levels (2-3 km), where the decrease in $J(NO_2)$ can be the same degree as those observed near the surface. This is because the aerosol extinction effect depends on particle size distribution. Fine particles have a higher extinction effect than coarse ones (Molnár and Mészáros, 2001), and they can be transported to higher levels.

The reduction in radiation due to SSA not only impacts $J(NO_2)$ but also affects the photolysis rate of other photochemical species, including $J(O_3)$ (Fig. S7 and S8), which are crucial for OH radical production in the atmosphere. The spatial and seasonal distribution patterns of $J(O_3)$ reductions mirror those of $J(NO_2)$, highlighting a consistent influence across photochemical species. These changes are poised to affect the photochemical formation of OH and $O_3$, the implications of which will be explored in subsequent sections.

### 3.4 Impacts of SSA on $HO_x$ radical

     The Cl radicals released from SSA contribute to atmospheric oxidation similarly to OH radicals, catalyzing the conversion of VOCs into $HO_2$ radicals. Figures 7a and 7b illustrate the spatial distribution of the SSA-induced increases in $HO_2$ radicals near the ground across eastern China and adjacent oceanic areas, corresponding with the significant rise in Cl radicals. This increase

is primarily due to the enhanced VOC degradation by Cl radicals. In remote oceanic regions, where VOC concentration is generally low, a decrease in $HO_2$ can be observed. This decrease is mainly attributed to the reduced photolysis rates due to the extinction effect of SSA, which seems to have countered any increases in $HO_2$ that would be caused by additionally available Cl radicals (Fig. 4c).

Figures 7c, 7d, and 7e present the vertical-diurnal variations of SSA-induced $HO_2$ concentrations in different months. Notably, $HO_2$ concentrations significantly increase after sunrise, driven by Cl radicals generated from the photolysis of ClNO and $ClNO_2$. However, a reduction in $HO_2$ is observed around noon, particularly at altitudes above the boundary layer, attributed to the pronounced radiative effect of SSA. In October, unlike other months, there is an increase in $HO_2$ concentration over the boundary layer in Beijing, suggesting a reduced radiative effect of SSA during this period in northern inland regions.

Moreover, a more substantial chemical contribution from Cl radicals in October leads to a sustained increase in $HO_2$ levels over the boundary layer.

     In the presence of NO, $HO_2$ converts into OH radicals, forming a critical $HO_x$ chemical cycle. Figures 8a and 8b show the spatial distribution of SSA-induced OH radicals near the ground, showing similar spatial and seasonal patterns to those of $HO_2$,

especially prominent in the North China Plain, Bohai Bay, and the YRD. However, in southern China, the area showing a

decrease in OH is more extensive than that for $HO_2$, stretching from oceanic regions to inland areas.

Figure 8 also shows the vertical-diurnal variations of SSA-induced OH concentrations in different months. The radiation effect distinctly influences OH concentrations, particularly in southern coastal cities. For instance, in July, the increase in OH concentration during the early morning is significantly offset by a noon-time decrease due to the radiation effect of SSA, resulting in an overall reduction in OH levels.

We employed the IRR process analysis module to elucidate the primary mechanisms driving the increase in OH radicals caused by SSA (Fig. 9). OH production is facilitated by several processes, including $HO_2$ conversion, $O_3$ photolysis, HONO photolysis, ozonolysis of some VOCs, and others such as $H_2O_2$ photolysis. Among these, $HO_2$ conversion accounts for over 70% of total OH generation, followed by $O_3$ and HONO photolysis. In Guangzhou, a southern coastal city, the contribution from $O_3$ photolysis to OH production is notably higher than that in other regions, likely due to the lower latitude with higher radiation levels. During the morning, the increase in OH is driven by enhanced $HO_2$ conversion caused by SSA. While in Guangzhou, the afternoon sees a decrease in $HO_2$ conversions, which can be attributed to the decreased $HO_2$ concentrations (Fig. 7e). Additionally, the reduced $O_3$ photolysis, exacerbated by SSA's enhanced radiative effects and an overall reduction in $O_3$ concentrations due to decreased $NO_x$ levels in a $NO_x$-limited regime (see section 3.5), further decreases OH production.

### 3.5 Impacts of SSA on $O_3$

Figure 10 illustrates the spatial distribution of changes in the maximum daily average 8-hour (MDA8) $O_3$ mixing ratio near the ground after incorporating SSA emissions into the model. In January, there is a notable increase in $O_3$ mixing ratios across eastern China and surrounding oceanic areas, peaking at 6.3 ppbv. In April, the North China Plain, Bohai Bay, and the Yellow Sea see increases up to 3.6 ppbv, while decreases up to 2.2 ppbv in southern China, South China Sea, and East China Sea. By July, increased $O_3$ levels are confined to smaller areas like Bohai Bay and the Yellow Sea, with a maximum rise of 3.6 ppbv, whereas decreases up to 1.8 ppbv are observed in the eastern coastal and oceanic regions of China. The variations of $O_3$ mixing ratios in October range from decreases of 2.8 ppbv to increases of 3.7 ppbv, showing similar patterns to April. We hypothesize that the regional and seasonal variations in $O_3$ largely depend on $O_3$ formation regime.

To determine the $O_3$ formation regime, we analyzed the ratio of production rates between hydrogen peroxide ($H_2O_2$) and $HNO_3$ ($P_{H2O2}/P_{HNO3}$) (Fig. 10c), using thresholds established in previous studies (Tonnesen and Dennis, 2000; Gaubert et al., 2021; Liu et al., 2021). A $P_{H2O2}/P_{HNO3}$ ratio below 0.06 indicates a VOC-limited region; ratios of 0.06 to less than 0.2 signify a

transition zone; and ratios of 0.2 or higher indicate a $NO_x$-limited region. Our results indicate that areas exhibiting increased MDA8 $O_3$ are primarily within VOC-limited regions, while decreases predominantly occur in $NO_x$-limited regions. In high $NO_x$ environments, the reaction between particulate $Cl^-$ from SSA and $NO_x$ leads to the formation of more Cl radicals, which can either reduce NO titration to $O_3$ or enhance $O_3$ production through interactions with VOCs. In contrast, in low $NO_x$ conditions, the reactions between particulate $Cl^-$ and $NO_x$ consume $NO_x$, resulting in lower $O_3$ levels. However, deviations from these patterns occur. For example, significant decreases in $O_3$ are observed in the PRD region during April, July, and October, and most of the regions in continental eastern China during summer, which are typically characterized as VOCs-limited. This phenomenon is likely due to SSA-induced reductions in $NO_2$ photolysis rates overshadowing potential increases in $O_3$ levels.

Figure 11 shows the vertical-diurnal variations in SSA-induced $O_3$ concentrations in Beijing, Shanghai, and Guangzhou. Morning increases within the planetary boundary layer across these cities are attributed to enhanced VOCs oxidation by SSA-induced Cl radicals, promoting the formation of $RO_2$ and $HO_2$ radicals that react with NO to generate $NO_2$, thereby increasing $O_3$ production. Moreover, the variations in other timeframes largely reflect the interplay between heterogeneous reactions and radiative effects. In January, the prevailing reduction in $NO_x$ within VOCs-limited areas elevates OH and $O_3$ levels in all three cities. April and October show similar patterns in Beijing and Shanghai, while Guangzhou exhibits declines due to the $NO_x$ reduction in a $NO_x$-limited environment coupled with decreased photolysis rates. In July, only in Beijing do $O_3$ levels rise during the morning hours, with reductions noted elsewhere, driven by $NO_x$ decreases in $NO_x$-limited conditions and reduced photolysis.

We also find significant decreases in SSA-induced $O_3$ concentrations over oceanic regions (Fig. 10) and in the upper levels (Fig. 11). This decline can be explained by two reasons: For one thing, remote oceanic areas (Fig. 10c) and upper levels (Wang et al., 2025; Lin et al., 2022) are generally in $NO_x$-limited conditions due to lower $NO_x$ concentrations, and the SSA-induced decrease in $NO_x$ (Fig. 3) reduce $O_3$ formation; for another, in these areas with scant VOCs, SSA-induced Cl radicals preferentially react with $O_3$ to form ClO (as depicted in Fig. S9 and S10), which enhances $O_3$ depletions. This behavior mirrors stratospheric conditions where Cl radicals are pivotal in consuming $O_3$.

Our analysis indicates that while SSA can suppress daytime $O_3$ and $HO_x$ levels through reduced photolysis rates, it also contributes to their morning production via the release of Cl radicals through heterogeneous reactions. Additionally, the interactions of SSA with nitrogen-containing species modulate $NO_x$ levels, affecting $O_3$ variations according to the prevailing formation regime, leading to regional and seasonal discrepancies in $O_3$ responses. The findings of this study contrast with previous modeling studies (Knipping and Dabdub, 2003; Dai et al., 2020; Sarwar and Bhave, 2007), which primarily reported

increases in $O_3$ attributable to SSA, highlighting the complex and variable impacts of SSA on coastal atmospheric chemistry.

**4 Conclusions and implications**

In this study, we utilized the WRF-CMAQ model to comprehensively investigate the complex interactions between SSA and continental anthropogenic emissions affecting $O_3$ formation in Eastern China. Figure 12 illustrates the mechanisms by which SSA influences radicals and $O_3$ formation in coastal areas. The process begins with the emission of SSA over oceanic areas.

In addition to its horizontal transport from the ocean to inland areas near surfaces, SSA is also transported extensively over continental regions through a long-range transport above the planetary boundary layer. Once inland, SSA interacts with pollutants from both continental anthropogenic and natural sources. Three primary pathways are identified by which SSA impacts radicals and $O_3$ formations: (1) SSA scatters solar radiation, reducing the photolysis rates of atmospheric chemicals and suppressing the daytime formation of $O_3$ and subsequently OH radicals. (2) Heterogeneous reactions between particulate

$Cl^-$ in SSA and nitrogen-containing species ($NO_2$ and $N_2O_5$) produce ClNO and $ClNO_2$, which are key precursors of Cl radicals. These SSA-induced Cl radicals oxidize VOCs and produce more OH, enhancing atmospheric oxidation capacity and $O_3$ production during morning hours. (3) These reactions also reduce $NO_x$ concentrations, an essential $O_3$ precursor. The resultant $O_3$ changes depend on its formation regime, subsequently influencing OH variations.

In summary, the influence of SSA on photochemical $O_3$ formation via the combination of these three pathways is both complex and variable, changing with regions and seasons. In winter, SSA notably increases OH and $O_3$ levels in eastern China due to significant $NO_x$ reductions in VOCs-limited areas. In contrast, in spring and autumn, while similar increases are found in the North China Plain, southern China experiences decreases due to $NO_x$ reductions in $NO_x$-limited areas and reduced photolysis rates. In summer, $O_3$ increases are confined to areas around Bohai, with reductions noted in other regions driven by $NO_x$

reductions in $NO_x$-limited areas and decreased photolysis.

This study suggests that as global efforts intensify to control anthropogenic emissions, the natural contributions from sources like SSA are likely to play an increasingly significant role in regional air quality and climate dynamics. This underscores the necessity for atmospheric chemistry models to integrate the diverse and seasonally varying impacts of natural aerosols like

SSA to improve predictions of air quality and to devise more effective environmental management strategies. This integration is crucial for accurately assessing future air quality trends and making informed policy decisions in the face of changing global emissions patterns.

**Author contributions**

Q.F., and Y.M.L. initiated the research. Y.M.L. and Y.Y.H. designed the research framework. Y.M.L. conducted model simulations and drew the figures. Y.M.L. and Y.Y.H. analyzed the results and wrote the paper with input from all authors. All authors contributed to the discussion and improvement of the paper.

**Financial support**

This research has been supported by the National Key Research and Development Program of China (2023YFC3710900), National Natural Science Foundation of China (42105097, 42375182), Science and Technology Program of Guangdong Province (Science and Technology Innovation Platform Category) (2019B121201002), Guangdong Basic and Applied Basic Research Foundation (2023A1515010162), and Guangdong Meteorological Bureau project (GRMC2024M07).

**Competing interests**

The authors declare that they have no conflict of interest.

**Code/Data availability**

The code or data used in this study are available upon request from Yiming Liu (liuym88@mail.sysu.edu.cn) and Yingying

Hong (yyhong0809@foxmail.com).

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

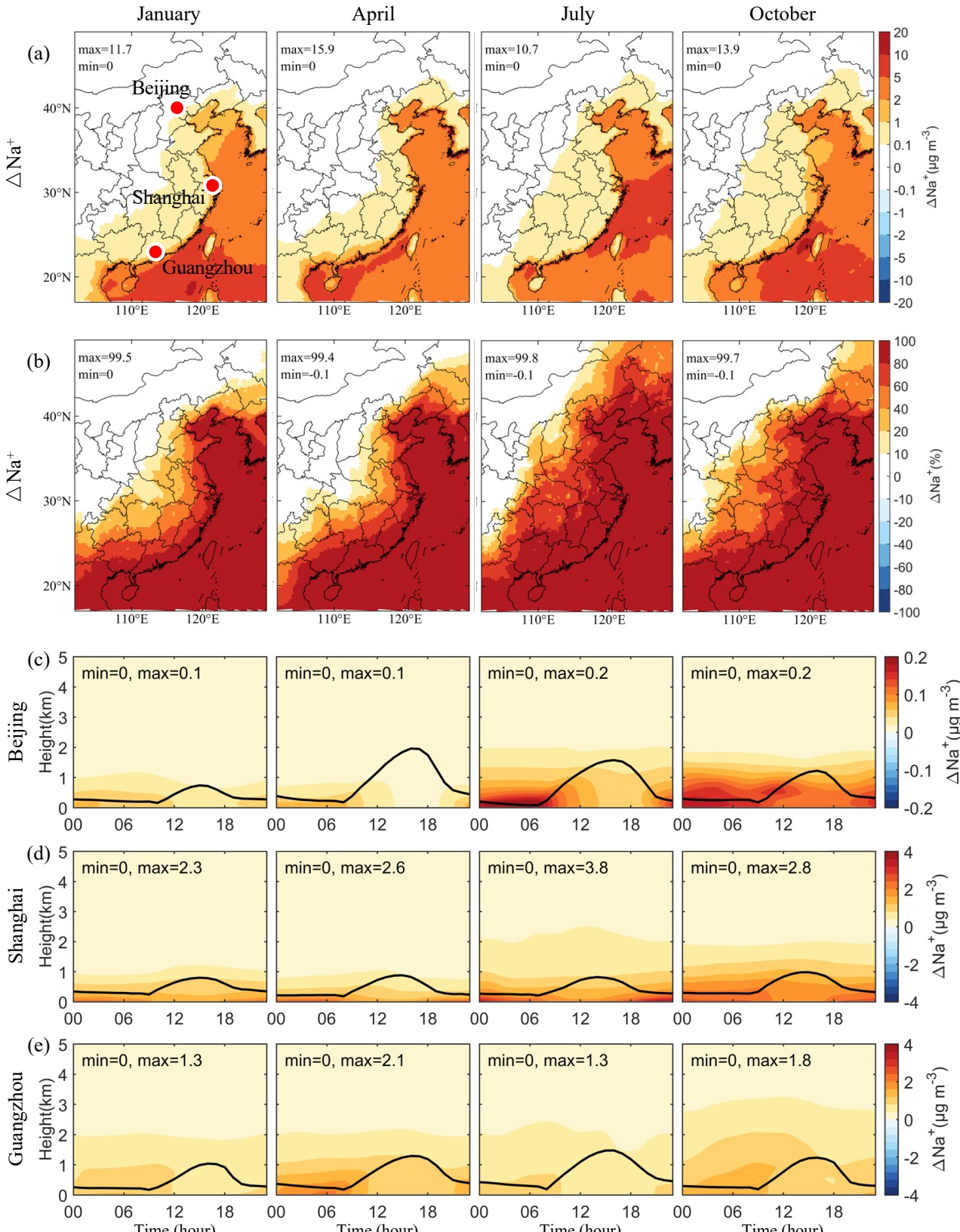

**Figure 1: Changes in simulated monthly mean concentrations of particulate Na$^+$ induced by SSA (BASE minus NOSA) during January, April, July, and October 2015. Panels (a) and (b) present the spatial distribution of changes and percentage changes, respectively. Panels (c-e) display the vertical-diurnal variations of changes in Beijing, Shanghai, and Guangzhou, respectively. The black line in Panels (c-e) is the simulated planetary boundary layer height.**

680

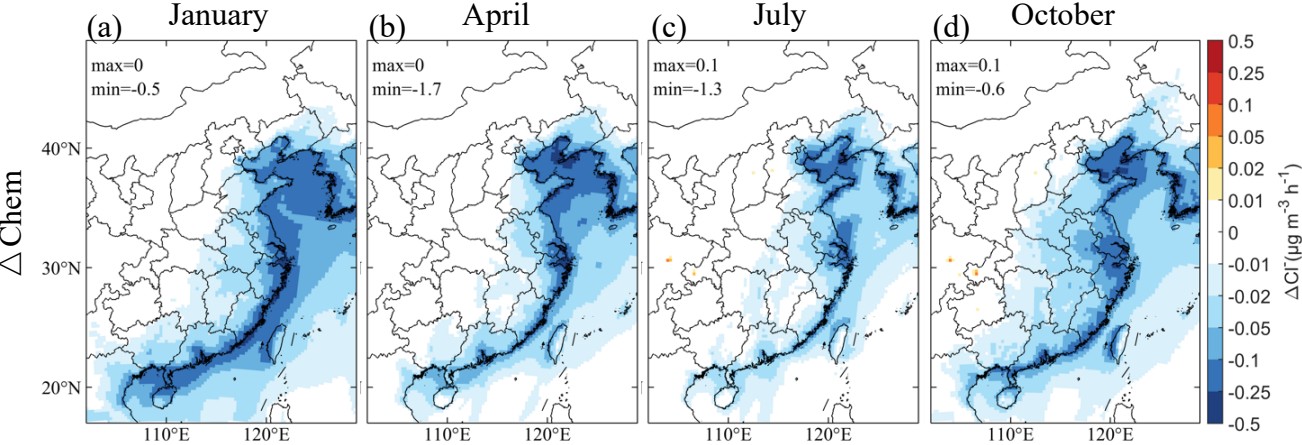

**Figure 2: Changes in the contributions of heterogeneous reactions to the simulated monthly mean concentrations of particulate Cl⁻ near the surface caused by SSA (BASE minus NOSA) during (a) January, (b) April, (c) July, and (d) October 2015. Heterogeneous reactions include reactions of particulate Cl⁻ with NO₂, NO₃, and N₂O₅.**

685

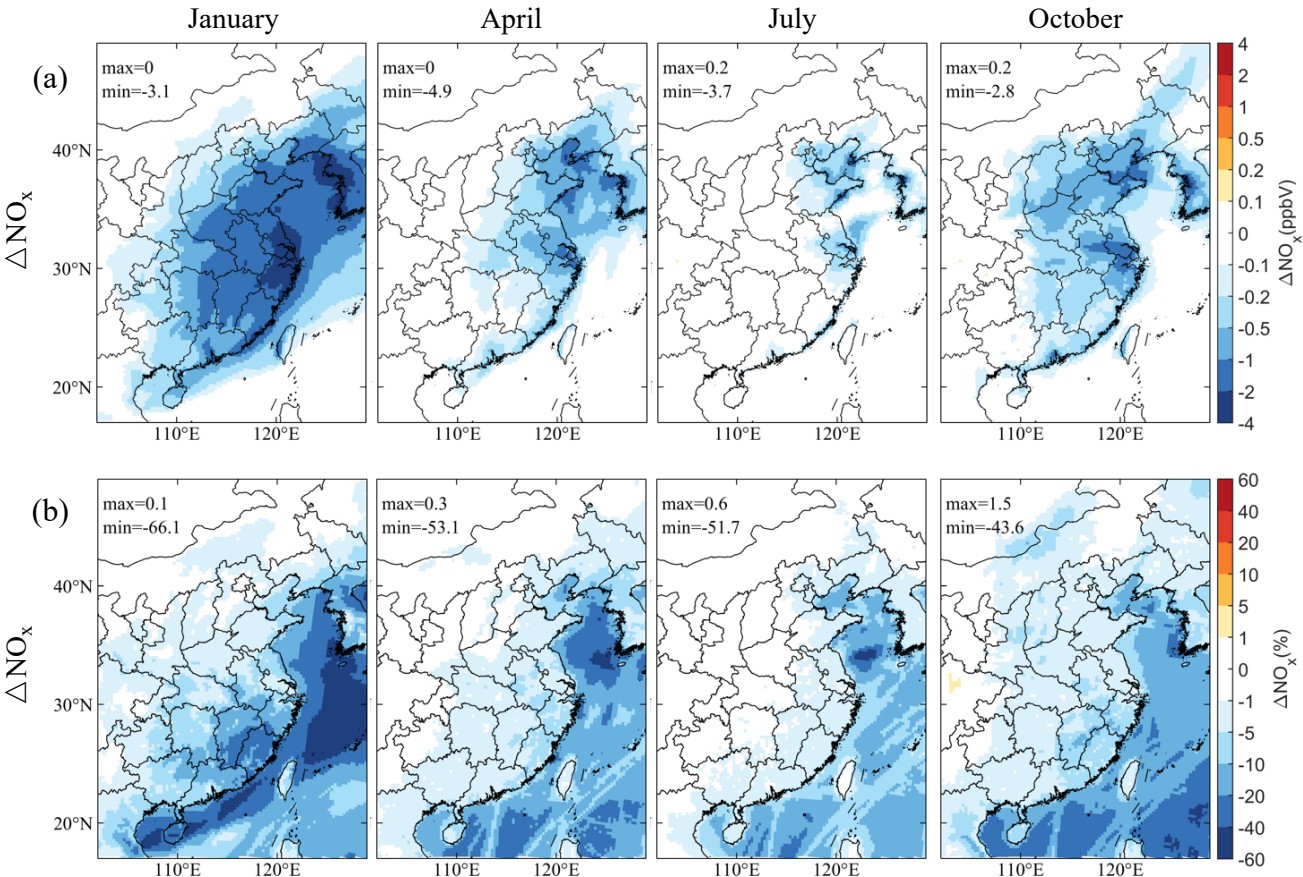

**Figure 3: Changes in simulated monthly mean NO$_x$ mixing ratios near the surface caused by SSA (BASE minus NOSA) during January, April, July, and October 2015. Panels (a) and (b) present the spatial distribution of changes and percentage changes, respectively.**

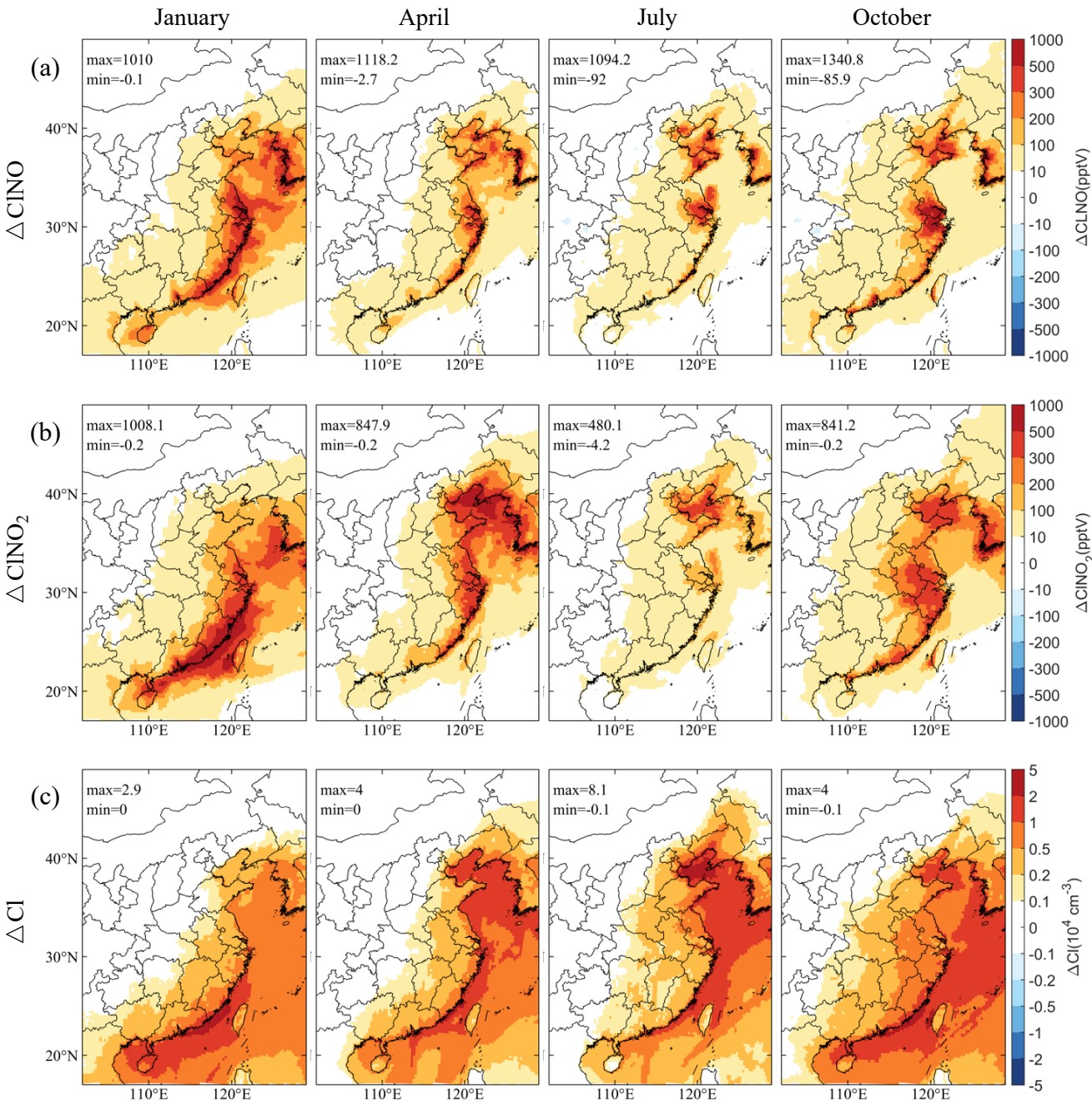

**Figure 4: Spatial distribution of changes in simulated monthly mean concentrations of (a) ClNO and (b) ClNO₂ at 5:00 LST, and (c) daily mean Cl radicals near the surface caused by SSA (BASE minus NOSA) during January, April, July, and October 2015.**

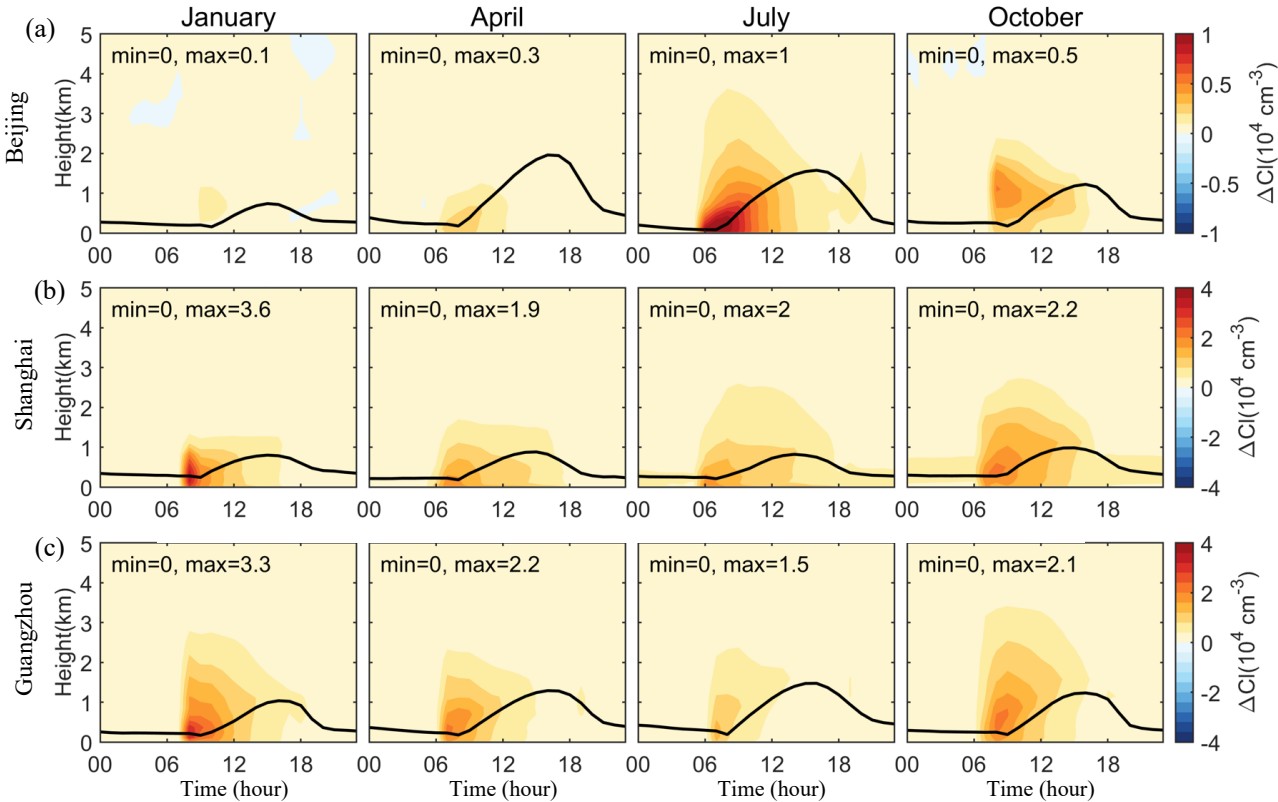

**Figure 5: Vertical-diurnal variations of changes in simulated monthly mean concentrations of Cl radicals caused by SSA (BASE minus NOSA) in (a) Beijing, (b) Shanghai, and (c) Guangzhou during January, April, July, and October 2015. The black line is the simulated planetary boundary layer height.**

700

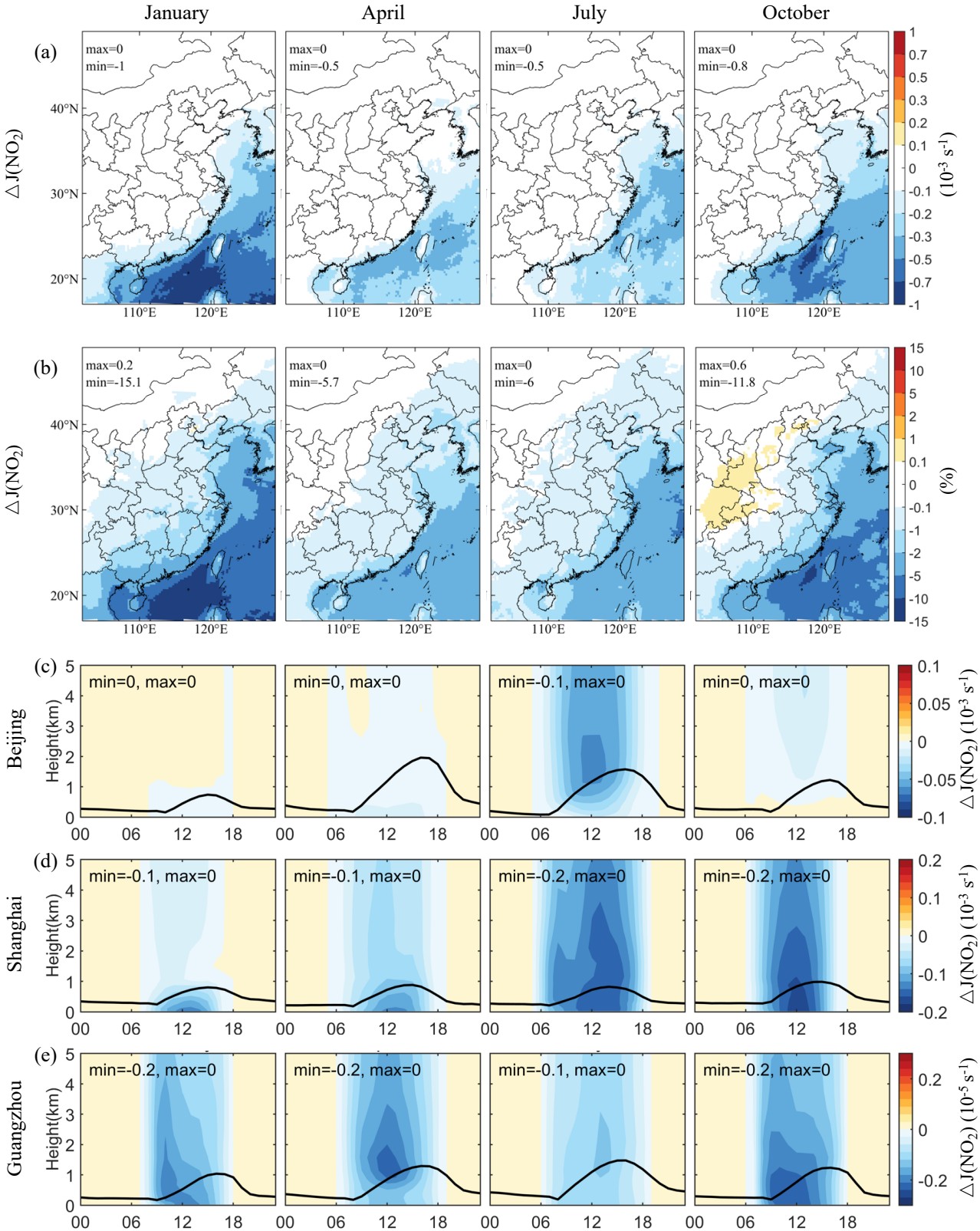

Figure 6: Changes in simulated monthly mean photolysis rate of NO$_2$ (J(NO$_2$)) induced by SSA (BASE minus NOSA) during January, April, July, and October 2015. Panels (a) and (b) present the spatial distribution of changes and percentage changes at 12:00 LST, respectively. Panels (c-e) display the vertical-diurnal variations of changes in Beijing, Shanghai, and Guangzhou, respectively. The black line in Panels (c-e) is the simulated planetary boundary layer height.

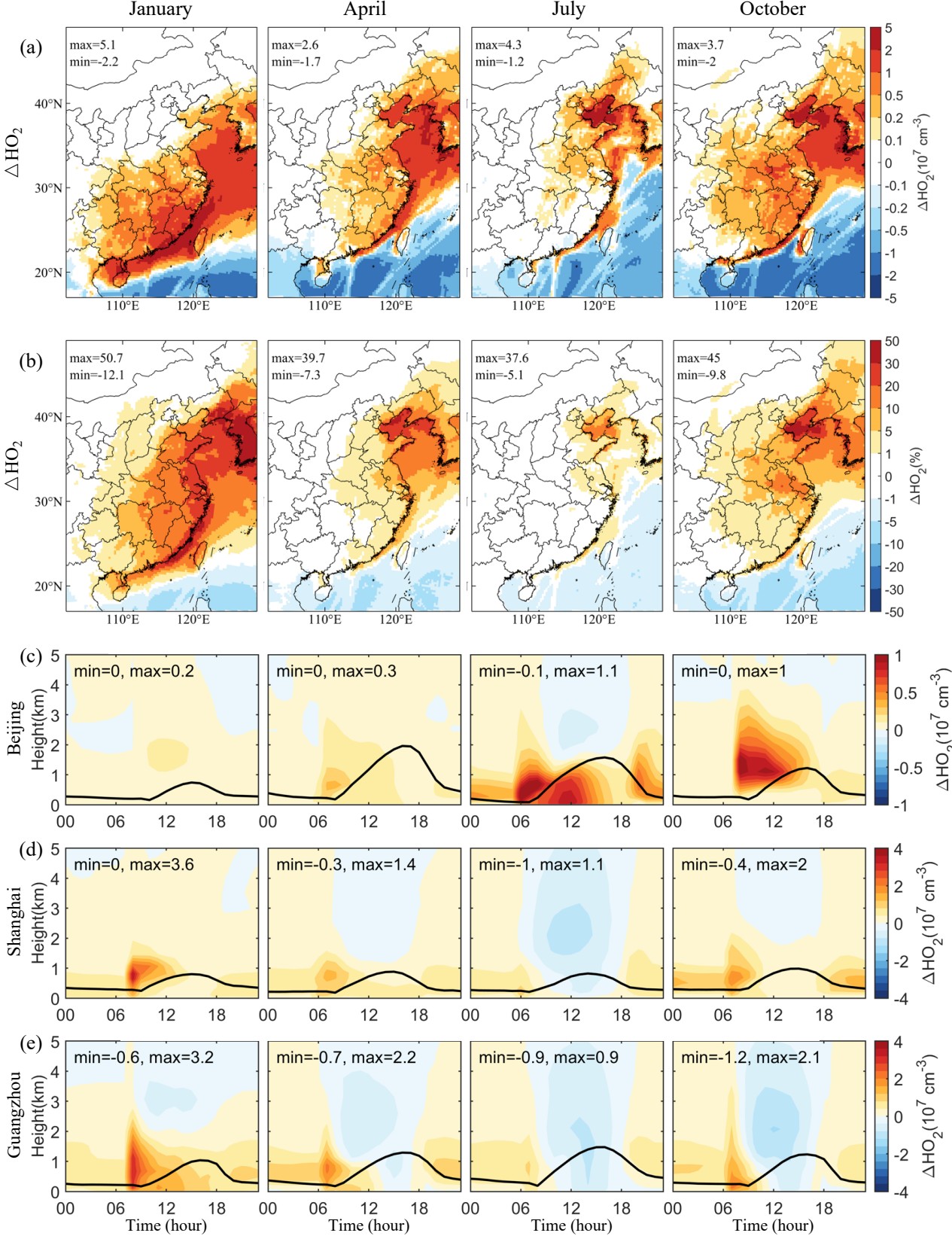

710

**Figure 7: The same as Fig. 1 but for simulated monthly mean HO₂ radical concentrations.**

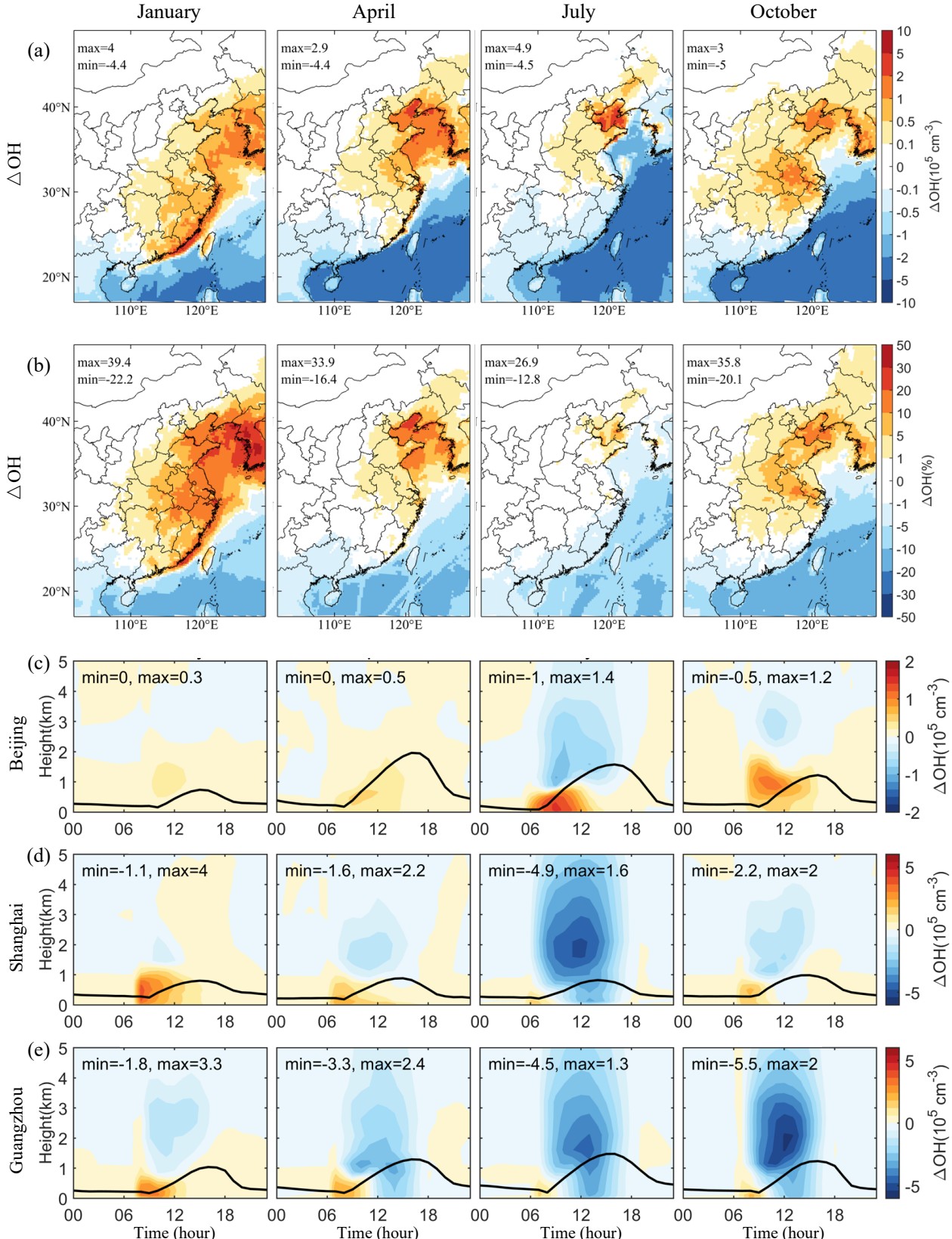

Figure 8: The same as Fig. 1 but for simulated monthly mean OH radical concentrations.

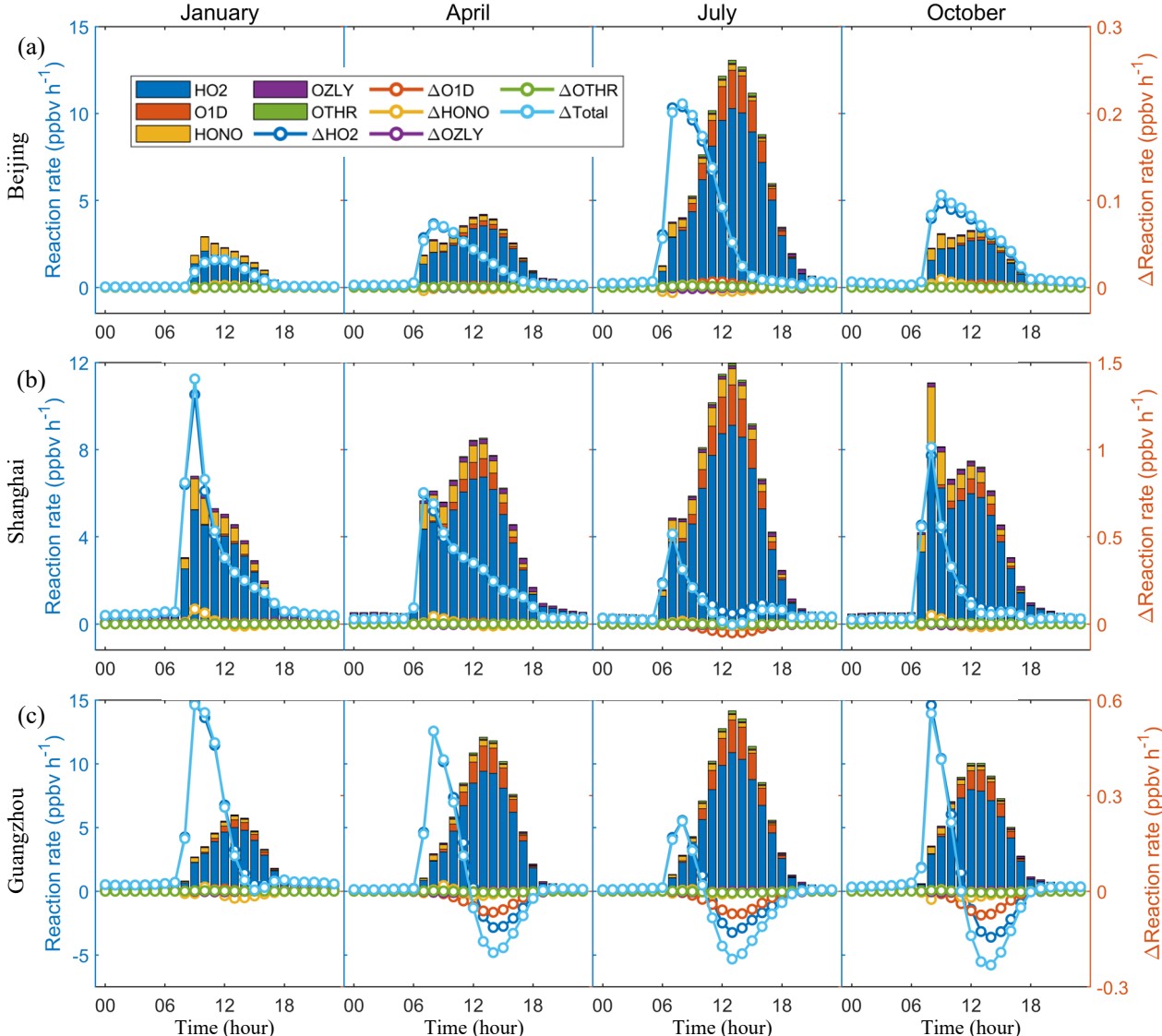

**Figure 9: Contribution of different pathways to the OH production near the surface and its changes caused by SSA (BASE minus NOSA) in (a) Beijing, (b) Shanghai, and (c) Guangzhou during January, April, July, and October 2015. These pathways include HO₂ conversion (HO₂), O₃ photolysis (O1D), HONO photolysis (HONO), ozonolysis of some VOCs (OZLY), and others (OTHR, including H₂O₂ photolysis, etc.).**

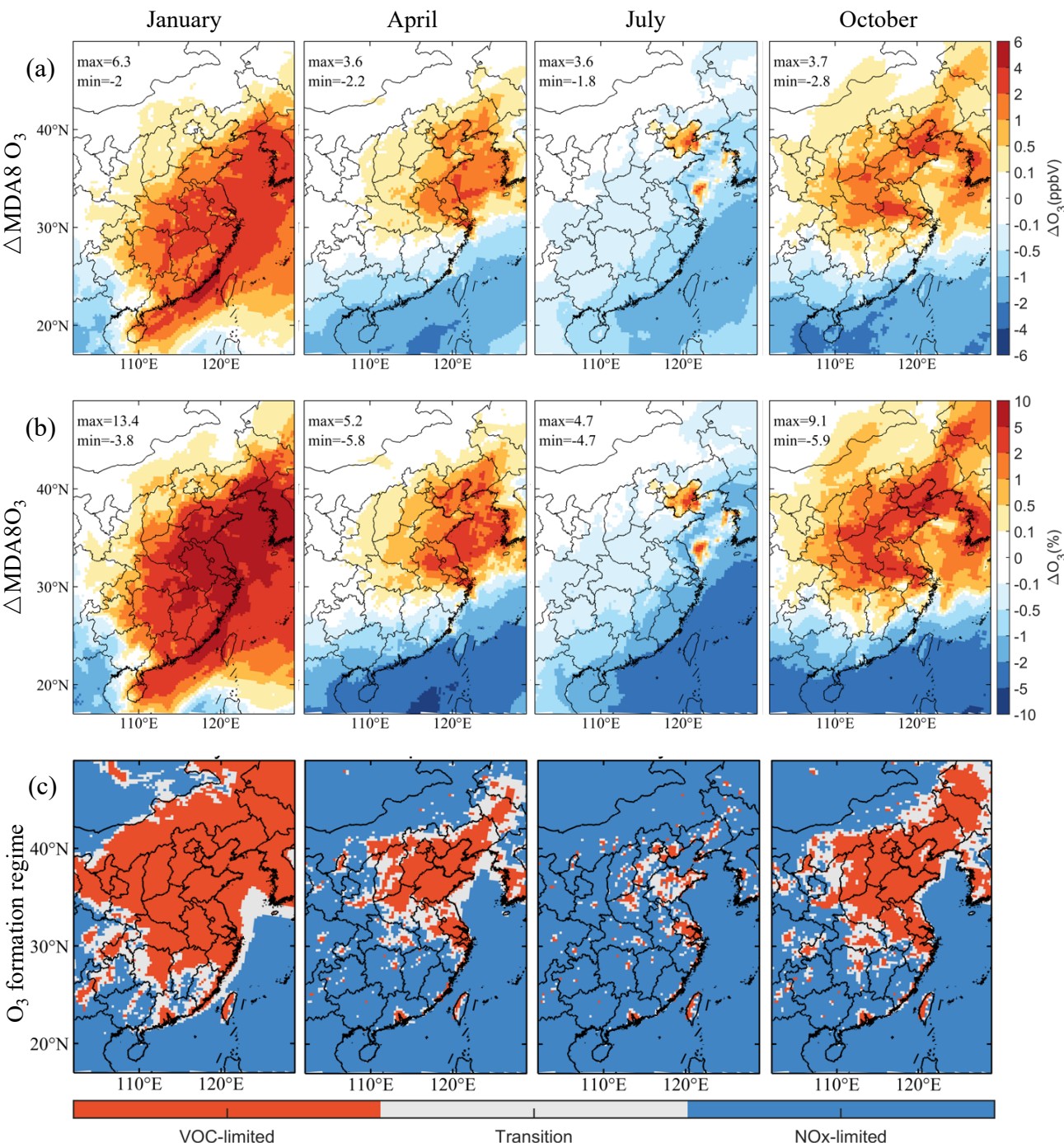

**Figure 10: Changes in the simulated monthly mean mixing ratio of MDA8 O₃ induced by SSA (BASE minus NOSA) and daytime O₃ formation regime during January, April, July, and October 2015. Panels (a) and (b) present the spatial distribution of changes and percentage changes, respectively. Panels (c) display the spatial distribution of the daytime (8:00 – 20:00 LST) O₃ formation regime. The regime is estimated by the ratio of the production rates of H₂O₂ to HNO₃ (P_{H2O2}/P_{HNO3}). VOC-limited region: P_{H2O2}/P_{HNO3} < 0.06; NOx-limited region: P_{H2O2}/P_{HNO3} ≥ 0.2, Transition zone: 0.06 ≤ P_{H2O2}/P_{HNO3} < 0.2. The production rates of H₂O₂ and HNO₃ are calculated using the integrated reaction rate (IRR) diagnose tool in the CMAQ model.**

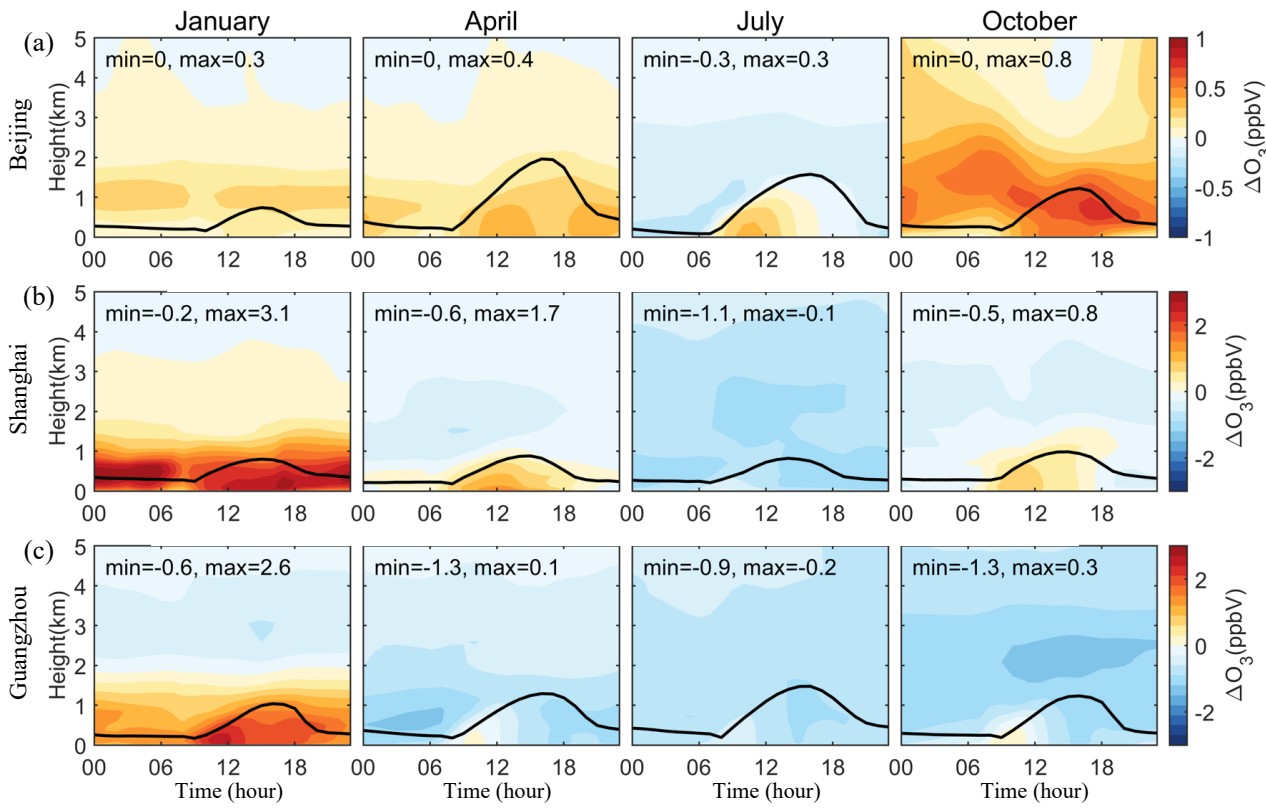

**Figure 11: Vertical-diurnal variations of changes in simulated monthly mean O₃ mixing ratios caused by SSA (BASE minus NOSA) in (a) Beijing, (b) Shanghai, and (c) Guangzhou during January, April, July, and October 2015. The black line is the simulated planetary boundary layer height.**

735

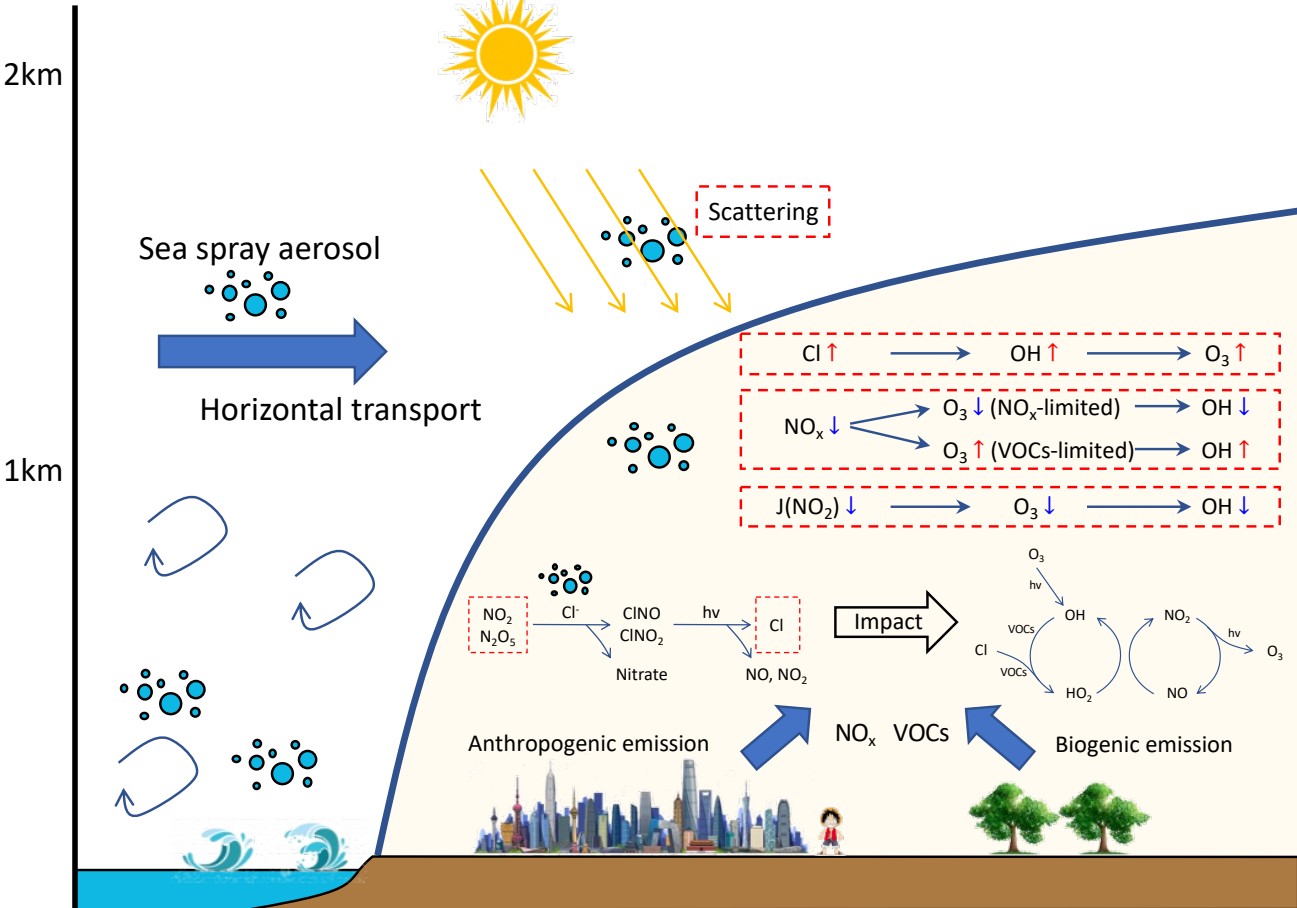

**Figure 12: Schematic map showing the impact of SSA on the radicals and O₃ formation in coastal areas.**