# Peer review of "The impact of sea spray aerosol on photochemical ozone formation over eastern China: heterogeneous reaction of chlorine particles and radiative effect"

_EGUsphere, 2024_

## Author Comment (AC1)

Response to the comment from Anonymous referee #2

The manuscript "The impact of sea spray aerosol on photochemical ozone formation over eastern China: heterogeneous reaction of chlorine particles and radiative effect", focuses on investigating the link between sea spray aerosol (SSA) and ozone in the atmosphere by employing the WFR-CMAQ model. To my mind the manuscript is well written and I believe that it is interesting to the scientific community and thus I recommend the paper for publication after the following comments have been addressed:

General comments:

[Comment 1] I find the information on the CMAQ model very short, no introduction is given at the beginning. I could also not find the full name of CMAQ, only the acronym. Please write a short paragraph describing the model and its general use.

Response: Thank you for your valuable suggestion. We have provided a more detailed description about the CMAQ model and its general use in the updated manuscript. We also give the full name of CMAQ.

Revision in the manuscript:

(1) Line 97: "Sarwar and Bhave (2007) utilized the Community Multiscale Air Quality Modeling System (CMAQ) model to explore …"

(2) Line 114: "Here we used the WRF-CMAQ model to perform air quality simulations in this study. The CMAQ (version 5.1) model is a regional chemical transport model developed by the United States Environmental Protection Agency (Appel et al., 2017). It has been widely used to explore the mechanism of multiple air quality issues, including tropospheric ozone, fine particles, acid deposition, and visibility degradation (Zhu et al., 2024; Kitagawa et al., 2021; Onwukwe and Jackson, 2021). The meteorological inputs of CMAQ model (version 5.1) were provided by the Weather Research and Forecasting (WRF) model."

[Comment 2] Regarding the aspect of humidity for the reactions studied (briefly mentioned on page 5), what kind of deliquescence and efflorescence behavior was used for SSA? Was it chosen size specific? Was the composition of dry SSA chosen to be the same for all sizes?

Response: Thank you for your valuable suggestion. The AERO6i aerosol module in CMAQ employs ISORROPIA (Binkowski and Roselle, 2003; Fountoukis and Nenes, 2007; Kelly et al., 2010) to simulates inorganic aerosol thermodynamics. The ISORROPIA model automatically adjusts the liquid/solid phase state of aerosols based on ambient relative humidity, thereby implicitly accounting for the effects of deliquescence and efflorescence. CMAQ does not explicitly differentiate the phase equilibrium processes between SSA and other soluble aerosols. Instead, they are uniformly handled by the thermodynamic model after accounting for their emissions (including SSA). Regarding heterogeneous chemical reactions involving chloride-containing particles, the uptake coefficients for $NO_2$ and $NO_3$ are assumed to be identical across different aerosol modes (Aitken, accumulation, and coarse). Additionally, the model assumes that the composition of emitted dry SSA is consistent across all sizes and does not vary with particle size. We have clarified the description in the manuscript.

Revision in the manuscript:

(1) Line 141: "The AERO6i aerosol module employed ISORROPIA (Binkowski and Roselle, 2003; Fountoukis and Nenes, 2007; Kelly et al., 2010) to uniformly simulates inorganic aerosol (including SSA) thermodynamics."

(2) Line 149: "This parameterization was identical across different aerosol modes (Aitken, accumulation, and coarse)."

(3) Line 179: "The composition of dry SSA in different aerosol modes remains consistent with that of seawater"

[Comment 3] In section 3.1, 3rd paragraph, $Cl^-$ is discussed and it is stated that Cl- emissions are higher compared to those of $Na^+$. Could you please specify why this is the case?

Response: Thank you for your valuable suggestion. We demonstrated in section 2.2 that the composition of dry SSA remains consistent with that of seawater, containing $Cl^-$ (55.4%), $Na^+$ (30.8%), $SO_4^{2-}$ (7.7%), $Mg^{2+}$ (3.8%), $Ca^{2+}$ (1.2%), and $K^+$ (1.1%). As a result, the percentage of particulate $Cl^-$ is higher than those of particulate $Na^+$, making the emission of particulate $Cl^-$ higher than those of $Na^+$. We have clarified the description here.

Revision in the updated manuscript:

(1) Line 222: "However, due to higher composition of particulate $Cl^-$ than $Na^+$ in SSA emissions, regions experiencing >80% change in $Cl^-$ are more extensive, underlining its substantial impact."

Specific comments:

[Comment 4] Page 2, line 42: delete "s" in "investigations"

Response: Corrected.

[Comment 5] Page 3, Eq. R1: please introduce the meaning of "cd"

Response: "cd" means the condensed phase. We have introduced it in Line 64.

[Comment 6] Page 4, line 108: add "s" to "demonstrate"

Response: Corrected.

[Comment 7] Page 7, line 197: delete "s" in "illustrates"

Response: Corrected.

[Comment 8] Page 9, line 260: The start of the sentence needs to be changed. I think you are referring to Fig. 5

Response: Thanks for pointing out this typo. We mistakenly removed some words of this paragraph in the original manuscript. We have corrected this issue.

Revision in the manuscript:
(1) Line 277: "Furthermore, the impact of SSA on Cl radicals is observed not only at the surface but also vertically through the atmosphere. Figure 5 examines the vertical-diurnal variations in SSA-induced Cl radical concentrations in Beijing, Shanghai, and Guangzhou."

[Comment 9] Page 10, line 290: delete "s" in "illustrates"
Response: Corrected.

[Comment 10] Page 10, line 304: change "present" to "presence". Delete "s" in "shows"
Response: Corrected.

[Comment 11] Page 12, line 343: please specify what "PRD region" is
Response: Thanks. We have defined PRD (Pearl River Delta) in Line 102.

[Comment 12] Page 13, line 372: add "s" to "illustrate"
Response: Corrected.

---

## Author Comment (AC2)

Response to the comment from Anonymous referee #1

"The impact of sea spray aerosol on photochemical ozone formation over eastern China: heterogeneous reaction of chlorine particles and radiative effect'"by Hong et al. is a well-written and well-motivated manuscript. It is of high interest and importance to detangle the influence of sea salt particles and their chloride-depleting reactions on ozone formation and the concentration of various species involved with the production of ozone. The manuscript is written clearly and concisely. I have a few questions/comments about the chemical reactions considered in the paper and the adjustment of photolysis rates due to increased scattering by aerosol particles. Those comments can be found below.

Specific scientific questions/comments:

[Comment 1] Lines 124 – 126: Have the abilities of the CMAQ model to adjust photolysis rates based on the presence of aerosols been verified or evaluated in previous works? Adjustments to photolysis rates are a big discussion point for the paper and its results, so it would be great to understand how accurate the estimated adjustments are in the first place.

Response: Thanks for your valuable suggestion. The calculation of photolysis rates in CMAQ uses an in-line approach for calculating actinic fluxes by solving a two-stream approximation of the radiative transfer equation (Binkowski et al., 2007; Toon et al., 1989) over wavebands based on the FAST-J photolysis model (Wild et al., 2000). Each layer includes scattering and extinction using simulated air density, cloud condensates, aerosols and trace gaseous such as $O_3$ and $NO_2$ (Appel et al., 2017). This approach has been verified or evaluated in some previous studies. Based on the aircraft measurement, Baker et al. (2018) found that the CMAQ model can well capture the observed $NO_2$ photolysis rates at ~2km height. Using this approach, Fu et al. (2014) concluded that the $NO_2$ and $O_3$ photolysis rates reduced by up to 2.4% and 1.9% respectively, due to the impact of dust aerosol during a heavy dust event. Fan and Li (2022) also found that the $O_3$ photolysis rates decreased by 1-4% due to the extinction effect of SSA. These references provide robustness of the CMAQ model to calculate the photolysis rates. It enables us to assess the effect of aerosols (e.g., SSA) on photochemical processes by adjusting photolysis rates accordingly. We have provided more discussions in the manuscript.

Revision in the updated manuscript:

(1) Line 129: "The calculation of photolysis rates in CMAQ uses an in-line approach for calculating actinic fluxes by solving a two-stream approximation of the radiative transfer equation (Binkowski et al., 2007; Toon et al., 1989) over wavebands based on the FAST-J photolysis model (Wild et al., 2000). Each layer includes scattering and extinction using simulated air density, cloud condensates, aerosols and trace gaseous such as $O_3$ and $NO_2$ (Appel et al., 2017). This approach has been verified or evaluated in some previous studies. Based on the aircraft measurement, Baker et al. (2018) found that the CMAQ model can well capture the observed $NO_2$ photolysis rates at ~2km height. Using this approach, Fu et al. (2014) concluded that the $NO_2$ and $O_3$ photolysis rates reduced by up to 2.4% and 1.9% respectively, due to the impact of dust aerosol during a heavy dust event. Fan and Li (2022) also found that the $O_3$ photolysis rates decreased by 1-4% due to the extinction effect of SSA. These references provide robustness of the CMAQ model to calculate the photolysis rates. It enables us to assess the effect of aerosols (e.g., SSA) on photochemical processes by adjusting photolysis rates

accordingly."

[Comment 2] Lines 128-1137: I see that the model's capability was not expanded to include reactions with SO₂ (g) and sea salt particles, which can be another source of chlorine radicals via chloride depletion. Would you mind adding some discussion here as to why SO₂ was not considered. I also wonder how it would affect the results and your comparisons to other studies if SO₂ were considered… some discussion on this would be nice.

Response: Thanks for raising this important issue. Gaseous $SO_2$ cannot react with SSA directly. However, it can be oxidized into $H_2SO_4$ and then react with SSA, releasing gaseous HCl. The oxidation of $SO_2$ is considered in the SAPRC07TIC gas-phase chemistry module and the reaction between $H_2SO_4$ and $Cl^-$ is handled in AERO6i aerosol module (ISORROPIA model) in original CMAQ. However, as shown in the following figure (Fig. R1), the changes of $SO_2$ mixing ratio induced by SSA is generally smaller than 0.1 ppbV, which suggested that the negligible role of $SO_2$ in the SSA impact. We have added some discussions in the revised manuscript.

[Figure]

Figure R1: Changes in simulated monthly mean $SO_2$ mixing ratios near surface caused by SSA during January, April, July and October 2015.

Revision in the updated manuscript:
(1) Line 141: "The AERO6i aerosol module employed ISORROPIA (Binkowski and Roselle, 2003; Fountoukis and Nenes, 2007; Kelly et al., 2010) to uniformly simulates inorganic aerosol thermodynamics. The chlorine deposition of SSA through its equilibrium reactions with $H_2SO_4$ and $HNO_3$ were considered in the model (Liu et al., 2015)"
(2) Line 283: "It should be noted that besides heterogenous reactions with nitrogen-containing species, particulate $Cl^-$ in SSA can react with $H_2SO_4$ and $HNO_3$ through thermodynamic equilibrium reactions, releasing gaseous HCl (Chi et al., 2015). HCl is another precursor of Cl radicals via its reaction with OH radicals, which generally occurred during daytime (Finlayson-Pitts, 2003). However, as shown in Fig. S6, the contribution of HCl to Cl radicals is much lower than the photolysis of ClNO and $ClNO_2$. Such small contribution of HCl were also reported in a box-model study in North China Plain (Liu et al., 2017). It suggests that the limited role of these thermodynamic equilibrium reactions in the Cl radicals and following $O_3$ formation."

[Comment 3] Lines 205-210: Just to confirm, when you are discussing particulate Cl⁻ here, is that the chlorine remaining after chloride depleting reactions have already been processed in the model? Or are you just discussing the simple change in particulate chloride concentrations before and after including them in the model (BASE vs. NOSA)? I ask because at first, I thought you were just showing the change in particulate Cl⁻ moving from NOSA to BASE, but before accounting for chloride depleting reactions. However, you mention that particulate chloride concentrations are higher aloft due to depletion reactions near the surface. This implies you have already run the model in full and accounted for depletion reactions when discussing these results. A bit of clarification would be very helpful here to know if 'particulate chloride' is referring to conditions before or after the depletion reactions have been accounted for by the model. The caption in figures S3 and S4 is not explicit in saying if these are particulate Cl⁻ mass concentrations *before or after* accounting for chloride depleting reactions. I was a bit confused as it seemed the discussion of chloride depletion really began in the subsequent section (Sect. 3.2) and that Sect 3.1 was more centered on discussing the results of considering sea salt particles in the model.

Response: Sorry for making this confusion. We just discuss the simple change in particulate Cl concentrations before and after including them in the model (BASE minus NOSA). We have clarified the description in the revised manuscript. The figure captions in the manuscript and SI are also clarified.

Revision in the updated manuscript:

(1) Line 221: "Additionally, the simulated changes in particulate Cl⁻ concentrations due to SSA (BASE minus NOSA) are shown in Fig S3 and S4."

(2) Line 225: "In polluted lower atmosphere, particulate Cl⁻ in SSA can be chemically depleted through thermodynamic equilibrium processes and heterogeneous reactions, which will be discussed in the following section."

[Comment 4] Line 207-208: You mention that depletion reactions with HNO₃ and H₂SO₄ may explain lower particulate Cl⁻ concentrations near the surface. I didn't think you were accounting for reactions with H₂SO₄ in the model, so how would reactions with H₂SO₄ be a partial explanation for the lower simulated Cl⁻ mass concentrations near the surface?

Response: Thanks for your comment. The original CMAQ model has considered the reactions of particulate Cl⁻ with HNO₃ and H₂SO₄. These thermodynamic equilibrium reactions are handled in the ISORROPIA model. Please also see our responses to the above 2ⁿᵈ comment. The chlorine depletion of SSA with H₂SO₄ and HNO₃ can be partial contributors to the depletion of SSA near the surface, but its impact on Cl radicals is limited (see Fig. S6). We have added some discussions in the manuscript.

Revision in the manuscript:

(1) Line 129: "The AERO6i aerosol module employed ISORROPIA (Binkowski and Roselle, 2003; Fountoukis and Nenes, 2007; Kelly et al., 2010) to uniformly simulates inorganic aerosol thermodynamics. The chlorine deposition of SSA through its equilibrium reactions with H2SO4 and HNO₃ were considered in the model (Liu et al., 2015)"

(2) Line 283: "It should be noted that besides heterogenous reactions with nitrogen-containing species, particulate Cl⁻ in SSA can react with H₂SO₄ and HNO₃ through thermodynamic equilibrium reactions, releasing gaseous HCl (Chi et al., 2015). HCl is another precursor of Cl radicals via its

reaction with OH radicals, which generally occurred during daytime (Finlayson-Pitts, 2003). However, as shown in Fig. S6, the contribution of HCl to Cl radicals is much lower than the photolysis of ClNO and ClNO$_2$. Such small contribution of HCl were also reported in a box-model study in North China Plain (Liu et al., 2017). It suggests that the limited role of these thermodynamic equilibrium reactions in the Cl radicals and following O$_3$ formation."

[Comment 5] Lines 223 – 228: Again, since you are providing specific numbers for changes in NO$_x$, I think it would be great to reiterate that you are not accounting for reactions between SO$_2$ and sea salt particles in your model simulations. If you have an idea of how significantly/insignificantly accounting for reactions with SO$_2$ and sea salt particles would affect your results, that may be good to mention here.

Response: Thanks for your valuable suggestion. According to our response to the 2$^{st}$ comment, the reactions between SO$_2$ and SSA in our model simulation is insignificant, which may not affect our results. We have provided some discussion in the revised manuscript.

Revision in the updated manuscript:

(1) Line 283: "It should be noted that besides heterogenous reactions with nitrogen-containing species, particulate Cl$^-$ in SSA can react with H$_2$SO$_4$ and HNO$_3$ through thermodynamic equilibrium reactions, releasing gaseous HCl (Chi et al., 2015). HCl is another precursor of Cl radicals via its reaction with OH radicals, which generally occurred during daytime (Finlayson-Pitts, 2003). However, as shown in Fig. S6, the contribution of HCl to Cl radicals is much lower than the photolysis of ClNO and ClNO$_2$. Such small contribution of HCl were also reported in a box-model study in North China Plain (Liu et al., 2017). It suggests that the limited role of these thermodynamic equilibrium reactions in the Cl radicals and following O$_3$ formation."

[Comment 6] Line 262: You mention increases in Cl radicals after sunrise are more pronounced at higher altitudes. Should there be something to direct readers to Fig. 5? I did not see Fig. 5 mentioned in the text, but it is possible I missed it. Can you be more clear by what you mean that 'increases in Cl radicals are more pronounced at higher altitudes shortly after sunrise'? In Fig. 5, I see the changes in Cl concentrations after sunrise are pretty similar in the boundary layer compared to right above the boundary layer for the three cities. Changes in Cl radical concentrations above 2 km are often 0.

Response: Thanks for your carefully review. This is a typo and we mistakenly removed some words of this paragraph in the original manuscript. In the revised manuscript, we have mentioned Fig. 5 in the text to explain the diurnal and vertical changes in Cl radicals induced by SSA. Besides, we have provided a clearer description and discussion in the vertical changes of Cl radicals.

Revision in the updated manuscript:

(1) Line 277: "Furthermore, the impact of SSA on Cl radicals is observed not only at the surface but also vertically through the atmosphere. Figure 5 examines the vertical-diurnal variations in SSA-induced Cl radical concentrations in Beijing, Shanghai, and Guangzhou."

(2) Line 280: "These increases are more pronounced near the top of planetary boundary layer shortly after sunrise, suggesting the impact of SSA on Cl radicals is more significant in upper levels than near the surface."

[Comment 7] Lines 275 – 286: It's interesting that J(NO$_2$) was decreased considerably in the upper troposphere. That high, you would presumably have higher actinic flux and less scattering from sea salt than at the surface. Fig 1 shows that changes in sea salt particle mass (using Na+ as a proxy) in the upper troposphere are close to zero. Are these low mass concentrations enough to scatter enough radiation to reduce J(NO$_2$) to the same degree as it is reduced at the surface, especially considering the higher actinic flux aloft? Perhaps some discussion here to elaborate on the result would be useful and of interest.

Response: Thanks for your valuable comment. The effect of aerosol particles on the extinction depends on their size distribution. Generally, fine particles have higher extinction effect than coarse particles (Molnár and Mészáros, 2001), and it can be transported to the higher levels. As a result, despite low concentrations in upper levels (2-3 km), fine SSA can reduce J(NO$_2$) to the same degree as those near the surface. We have provided some discussions in the revised manuscript.

Revision in the manuscript:

(1) Line 303: "We note that the extinction effect of SSA can extend into the upper levels (2-3 km), where the decrease in J(NO$_2$) can be the same degree as those observed near the surface. This is because the aerosol extinction effect depends on particle size distribution. Fine SSA particles have higher extinction effect than coarse ones (Molnár and Mészáros, 2001), and it can be transported to the higher levels."

[Comment 8] Lines 291 – 294: Correct me if I'm wrong but Fig. 7 shows differences between the NOSA and BASE simulations. Thus, wouldn't the VOC concentrations be the same over the remote oceanic regions in both scenarios? If not, please explain why. If not, then perhaps the main explanation for lower HO$_2$ concentrations over remote oceanic regions is reduced photolysis due to scattering by sea salt particles? It's impressive the reduction in photolysis due to light scattering is enough to decrease production of HO$_2$ when Cl-radicals from chloride depletion reactions are considered in the model. Fig. 4 shows that there were considerable increases in Cl- radicals over these remote oceanic regions, yet the competing effect of scattering by sea salt particles seems to have countered any increases in HO$_2$ that would be caused by additionally available Cl- radicals. As mentioned earlier, it would be great to understand and/or mention the robustness and accuracy of the changes in photolysis rates due to increased SSA scattering in the model since this is such a prominent topic/result of the paper.

Response: Thank you for your valuable suggestion. The VOC concentrations are the same over the remote oceanic regions in both scenarios. We agree that the decreased HO$_2$ over remote oceanic regions is mainly attributed to the reduced photolysis rates caused by SSA. Please see our responses to the 1$^{st}$ comment for the discussions of the CMAQ module to calculate the photolysis rates.

Revision in the manuscript:

(1) Line 319: "In remote oceanic regions, where VOC concentration is generally low, a decrease in HO$_2$ can be observed. This decrease is mainly attributed to the reduced photolysis rates due to the extinction effect of SSA, which seems to have countered any increases in HO$_2$ that would be caused by additionally available Cl radicals (Fig. 4c)."

[Comment 9] Lines 309-312: Similar comment as above. The decreases in OH concentrations in the troposphere are interesting considering that sea salt particle mass concentrations are presumably low at those altitudes and actinic fluxes are stronger. I know the transport of SSA is possible above the boundary layer as you mentioned, but changes in SSA above 3 km are mostly zero across all months. It is interesting that the scattering from such relatively low amounts of SSA is enough to offset any potential increases in OH due to increased presence of Cl- radicals (although the change in Cl- radicals above 3 km is also close to 0 for all months). There is not really an action item for this comment, I just found it interesting and I think the results would be strengthened by mentioning/citing the validity of the simulated changes in photolysis rates due to increased SSA scattering at least once somewhere in the paper.

Response: Thanks for your valuable suggestion. There are some previous studies verified and evaluated the module to calculate the photolysis rates in CMAQ. We have cited these references in the manuscript. Please also see our responses to the above 1st and 7th comment.

[Comment 10] Lines 335 – 356: The relationship between the sign of the change in $O_3$ and whether or not the region is NOx- or VOC-limited is interesting. Previously, reductions in concentrations of various radicals and species involved with the NOx-VOC-O3 system of reactions were attributed primarily to changes in photolysis rates due to scattering by sea salt particles. I wonder now if whether those altitudes are NOx- or VOC-limited may be of interest to consider for explaining reductions in the concentrations of those species at higher altitudes? The regime is mentioned when considering the vertical profile in changes of $O_3$ concentrations, so I wonder if a similar discussion of regime would be appropriate to at least mention for other species involved in reactions related to the production of $O_3$?

Response: Thank you for your valuable suggestion to further explore the role of $NO_x$/VOC-limited regimes in explaining altitude-dependent changes in ozone-related species. Higher levels are generally in $NO_x$-limited conditions due to lower $NO_x$ concentrations (Wang et al., 2025; Lin et al., 2022), and the SSA-induced decrease in $NO_x$ can reduce $O_3$ formation. We have added some discussions in the revised manuscript.

Revision in the manuscript:

(1) Line 386: "We also find significant decreases in SSA-induced $O_3$ concentrations over oceanic regions (Fig. 10) and in the upper levels (Fig. 11). This decline can be explained by two reasons: For one thing, remote oceanic areas (Fig. 10c) and upper levels (Wang et al., 2025; Lin et al., 2022) are generally in $NO_x$-limited conditions due to lower $NO_x$ concentrations, and the SSA-induced decrease in $NO_x$ (Fig. 3) reduce $O_3$ formation; for another, in these areas with scant VOCs, SSA-induced Cl radicals preferentially react with $O_3$ to form ClO (as depicted in Fig. S9 and S10), which enhances $O_3$ depletions. This behavior mirrors stratospheric conditions where Cl radicals are pivotal in consuming $O_3$."

[Comment 11] Line 260: Typo? Says "Furthe es"

Response: Thanks for pointing out this typo. We mistakenly removed some words of this paragraph in the original manuscript. We have corrected this issue.

Revision in the updated manuscript:

(1) Line 277: "Furthermore, the impact of SSA on Cl radicals is observed not only at the surface but also vertically through the atmosphere. Figure 5 examines the vertical-diurnal variations in SSA-induced Cl radical concentrations in Beijing, Shanghai, and Guangzhou."

---

## Referee Report (RR1)

Response to the comment from Anonymous referee #1

"The impact of sea spray aerosol on photochemical ozone formation over eastern China: heterogeneous reaction of chlorine particles and radiative effect'"by Hong et al. is a well-written and well-motivated manuscript. It is of high interest and importance to detangle the influence of sea salt particles and their chloride-depleting reactions on ozone formation and the concentration of various species involved with the production of ozone. The manuscript is written clearly and concisely. I have a few questions/comments about the chemical reactions considered in the paper and the adjustment of photolysis rates due to increased scattering by aerosol particles. Those comments can be found below.

Specific scientific questions/comments:

[Comment 1] Lines 124 – 126: Have the abilities of the CMAQ model to adjust photolysis rates based on the presence of aerosols been verified or evaluated in previous works? Adjustments to photolysis rates are a big discussion point for the paper and its results, so it would be great to understand how accurate the estimated adjustments are in the first place.

Response: Thanks for your valuable suggestion. The calculation of photolysis rates in CMAQ uses an in-line approach for calculating actinic fluxes by solving a two-stream approximation of the radiative transfer equation (Binkowski et al., 2007; Toon et al., 1989) over wavebands based on the FAST-J photolysis model (Wild et al., 2000). Each layer includes scattering and extinction using simulated air density, cloud condensates, aerosols and trace gaseous such as $O_3$ and $NO_2$ (Appel et al., 2017). This approach has been verified or evaluated in some previous studies. Based on the aircraft measurement, Baker et al. (2018) found that the CMAQ model can well capture the observed $NO_2$ photolysis rates at ~2km height. Using this approach, Fu et al. (2014) concluded that the $NO_2$ and $O_3$ photolysis rates reduced by up to 2.4% and 1.9% respectively, due to the impact of dust aerosol during a heavy dust event. Fan and Li (2022) also found that the $O_3$ photolysis rates decreased by 1-4% due to the extinction effect of SSA. These references provide robustness of the CMAQ model to calculate the photolysis rates. It enables us to assess the effect of aerosols (e.g., SSA) on photochemical processes by adjusting photolysis rates accordingly. We have provided more discussions in the manuscript.

Revision in the updated manuscript:

(1) Line 129: "The calculation of photolysis rates in CMAQ uses an in-line approach for calculating actinic fluxes by solving a two-stream approximation of the radiative transfer equation (Binkowski et al., 2007; Toon et al., 1989) over wavebands based on the FAST-J photolysis model (Wild et al., 2000). Each layer includes scattering and extinction using simulated air density, cloud condensates, aerosols and trace gaseous such as $O_3$ and $NO_2$ (Appel et al., 2017). This approach has been verified or evaluated in some previous studies. Based on the aircraft measurement, Baker et al. (2018) found that the CMAQ model can well capture the observed $NO_2$ photolysis rates at ~2km height. Using this approach, Fu et al. (2014) concluded that the $NO_2$ and $O_3$ photolysis rates reduced by up to 2.4% and 1.9% respectively, due to the impact of dust aerosol during a heavy dust event. Fan and Li (2022) also found that the $O_3$ photolysis rates decreased by 1-4% due to the extinction effect of SSA. These references provide robustness of the CMAQ model to calculate the photolysis rates. It enables us to assess the effect of aerosols (e.g., SSA) on photochemical processes by adjusting photolysis rates accordingly."

**Reviewer response: Thank you for adding this. The references you added are helpful.**

[Comment 2] Lines 128-1137: I see that the model's capability was not expanded to include reactions with $SO_2$ (g) and sea salt particles, which can be another source of chlorine radicals via chloride depletion. Would you mind adding some discussion here as to why $SO_2$ was not considered. I also wonder how it would affect the results and your comparisons to other studies if $SO_2$ were considered… some discussion on this would be nice.

Response: Thanks for raising this important issue. Gaseous $SO_2$ cannot react with SSA directly. However, it can be oxidized into $H_2SO_4$ and then react with SSA, releasing gaseous HCl. The

oxidation of $SO_2$ is considered in the SAPRC07TIC gas-phase chemistry module and the reaction between $H_2SO_4$ and $Cl^-$ is handled in AERO6i aerosol module (ISORROPIA model) in original CMAQ. However, as shown in the following figure (Fig. R1), the changes of $SO_2$ mixing ratio induced by SSA is generally smaller than 0.1 ppbV, which suggested that the negligible role of $SO_2$ in the SSA impact. We have added some discussions in the revised manuscript.
Figure R1: Changes in simulated monthly mean $SO_2$ mixing ratios near surface caused by SSA during January, April, July and October 2015.
Revision in the updated manuscript:
(1) Line 141: "The AERO6i aerosol module employed ISORROPIA (Binkowski and Roselle, 2003; Fountoukis and Nenes, 2007; Kelly et al., 2010) to uniformly simulates inorganic aerosol thermodynamics. The chlorine deposition of SSA through its equilibrium reactions with $H_2SO_4$ and $HNO_3$ were considered in the model (Liu et al., 2015)"
(2) Line 283: "It should be noted that besides heterogenous reactions with nitrogen-containing species, particulate $Cl^-$ in SSA can react with $H_2SO_4$ and $HNO_3$ through thermodynamic equilibrium reactions, releasing gaseous HCl (Chi et al., 2015). HCl is another precursor of Cl radicals via its reaction with OH radicals, which generally occurred during daytime (Finlayson-Pitts, 2003). However, as shown in Fig. S6, the contribution of HCl to Cl radicals is much lower than the photolysis of ClNO and $ClNO_2$. Such small contribution of HCl were also reported in a box-model study in North China Plain (Liu et al., 2017). It suggests that the limited role of these thermodynamic equilibrium reactions in the Cl radicals and following $O_3$ formation."

**Reviewer response: Thank you for addressing this. First, do you mean "chloride depletion of SSA through…" as opposed to 'chloride deposition of SSA through…"**

**I see that your model results show the production of Cl radicals from ClNO and ClNO2 to be more important than production from reactions involving H2SO4, HNO3, and SSA. If I understand correctly, reactions forming ClNO and ClNO2 would result in a replacement of particulate Cl- with NO3- in sea salt particles. Thus if pathways involving ClNO and ClNO2 are the dominant way in which reactive Cl in SSA is being released to the atmosphere, would you expect the composition of depleted SSA to contain mostly NO3- with relatively small amounts of SO4(2-)? I ask because studies looking at the composition of aged SSA such as Braun et al. (2017; https://doi.org/10.1021/acs.est.7b02039) and AzadiAghdam et al. (2019; https://doi.org/10.1016/j.atmosenv.2019.116922) show that a considerable amount of chloride depletion in submicron SSA can be attributed to SO4(2-) in certain conditions. Braun et al. is considering a different region, while AzadiAghdam et al. features data from the Philippines. I have seen other studies as well attributing non-negligible amounts of chloride depletion to reactions between H2SO4 and SSA when analyzing the composition of aged sea salt particles. Your study is looking at it from a different perspective (where you are NOT studying chloride depletion from the perspective of the chemical composition of the sea salt particles themselves), so I wonder how to resolve the differences between the perspectives when it comes to the role played by HNO3 and H2SO4. I certainly am not casting doubt on your model results, and I see they show changes in SO2 are small and that HCl is not a dominant contributor to Cl radicals.**

**I think researchers studying chloride depletion from the perspective of SSA chemical composition will find your results interesting regarding the chemical pathways leading to the presence of HNO3 in aged sea salt particles. I think currently many researchers see HNO3 in aged SSA and report that the HNO3 came to exist in the SSA through the pathway HNO3 + NaCl → HCl + NaNO3, whereas you show that other pathways involving N2O5 and NO2 may lead more dominantly to the presence of NaNO3 in SSA (if I understand your work correctly, please correct me if I'm misunderstanding).**

**For example, here is a quote from Su et al. (2022; https://doi.org/10.1016/j.atmosenv.2022.119365):**
**"The acid displacement reaction is regarded as a major chemical pathway in chloride depletion, releasing HCl:HA (g or aq) + NaCl (aq or s) →**
**NaA (aq or s) + HCl (g or aq) where HA (g or aq) represents acidic species (e.g., sulfuric acid,**

nitric acid, organic acid) in gaseous (*g*) or aqueous (*aq*) phases. NaCl $_{(aq \text{ or } s)}$ is the major component of SSA in aqueous or solid (*s*) phases. NaA is the sodium salt from the interaction between HA and NaCl. It should be noted that the acid displacement reaction involving organic acid in SSA is reversible."

I apologize for the long-winded comment. I just find it interesting that from the modeling perspective Cl radical formation from HCl is small compared to that from ClNO and ClNO2, whereas in sea salt chemistry papers, the main reaction presented in describing the process of Cl- depletion is HA + NaCl → NA + HCl.

This paper showcases how it is important to consider (1) modeling studies looking at the most important pathways regarding the production of Cl radicals and (2) studies examining chloride depletion from the perspective of SSA chemical composition as the two perspectives may be inclined to emphasize different reaction pathways in explaining the presence of HNO3 in aged SSA.

No action needed for this comment, just wanted to put these thoughts out there and see if you had any thoughts on this. I am more familiar with the particle chemistry side, so it would be interesting to hear your perspective from the modeling side.

[Comment 3] Lines 205-210: Just to confirm, when you are discussing particulate Cl- here, is that the chlorine remaining after chloride depleting reactions have already been processed in the model? Or are you just discussing the simple change in particulate chloride concentrations before and after including them in the model (BASE vs. NOSA)? I ask because at first, I thought you were just showing the change in particulate Cl- moving from NOSA to BASE, but before accounting for chloride depleting reactions. However, you mention that particulate chloride concentrations are higher aloft due to depletion reactions near the surface. This implies you have already run the model in full and accounted for depletion reactions when discussing these results. A bit of clarification would be very helpful here to know if 'particulate chloride' is referring to conditions before or after the depletion reactions have been accounted for by the model. The caption in figures S3 and S4 is not explicit in saying if these are particulate Cl- mass concentrations *before or after* accounting for chloride depleting reactions. I was a bit confused as it seemed the discussion of chloride depletion really began in the subsequent section (Sect. 3.2) and that Sect 3.1 was more centered on discussing the results of considering sea salt particles in the model.
Response: Sorry for making this confusion. We just discuss the simple change in particulate Cl concentrations before and after including them in the model (BASE minus NOSA). We have clarified the description in the revised manuscript. The figure captions in the manuscript and SI are also clarified.
Revision in the updated manuscript:
(1) Line 221: "Additionally, the simulated changes in particulate Cl- concentrations due to SSA (BASE minus NOSA) are shown in Fig S3 and S4."
(2) Line 225: "In polluted lower atmosphere, particulate Cl- in SSA can be chemically depleted through thermodynamic equilibrium processes and heterogeneous reactions, which will be discussed in the following section."

Reviewer response: Thank you for your response. I understand now!

[Comment 4] Line 207-208: You mention that depletion reactions with HNO3 and H2SO4 may explain lower particulate Cl- concentrations near the surface. I didn't think you were accounting for reactions with H2SO4 in the model, so how would reactions with H2SO4 be a partial explanation for the lower simulated Cl- mass concentrations near the surface?
Response: Thanks for your comment. The original CMAQ model has considered the reactions of

particulate Cl- with HNO₃ and H₂SO₄. These thermodynamic equilibrium reactions are handled in the ISORROPIA model. Please also see our responses to the above 2nd comment. The chlorine depletion of SSA with H₂SO₄ and HNO₃ can be partial contributors to the depletion of SSA near the surface, but its impact on Cl radicals is limited (see Fig. S6). We have added some discussions in the manuscript.

Revision in the manuscript:

(1) Line 129: "The AERO6i aerosol module employed ISORROPIA (Binkowski and Roselle, 2003; Fountoukis and Nenes, 2007; Kelly et al., 2010) to uniformly simulates inorganic aerosol thermodynamics. The chlorine deposition of SSA through its equilibrium reactions with H2SO4 and HNO₃ were considered in the model (Liu et al., 2015)"

(2) Line 283: "It should be noted that besides heterogenous reactions with nitrogen-containing species, particulate Cl- in SSA can react with H2SO4 and HNO₃ through thermodynamic equilibrium reactions, releasing gaseous HCl (Chi et al., 2015). HCl is another precursor of Cl radicals via its reaction with OH radicals, which generally occurred during daytime (Finlayson-Pitts, 2003). However, as shown in Fig. S6, the contribution of HCl to Cl radicals is much lower than the photolysis of ClNO and ClNO₂. Such small contribution of HCl were also reported in a box-model study in North China Plain (Liu et al., 2017). It suggests that the limited role of these thermodynamic equilibrium reactions in the Cl radicals and following O₃ formation."

**Reviewer response: Thank you for addressing this comment.**

[Comment 5] Lines 223 – 228: Again, since you are providing specific numbers for changes in NOₓ, I think it would be great to reiterate that you are not accounting for reactions between SO₂ and sea salt particles in your model simulations. If you have an idea of how significantly/insignificantly accounting for reactions with SO₂ and sea salt particles would affect your results, that may be good to mention here.

Response: Thanks for your valuable suggestion. According to our response to the 2st comment, the reactions between SO₂ and SSA in our model simulation is insignificant, which may not affect our results. We have provided some discussion in the revised manuscript.

Revision in the updated manuscript:

(1) Line 283: "It should be noted that besides heterogenous reactions with nitrogen-containing species, particulate Cl- in SSA can react with H2SO4 and HNO₃ through thermodynamic equilibrium reactions, releasing gaseous HCl (Chi et al., 2015). HCl is another precursor of Cl radicals via its reaction with OH radicals, which generally occurred during daytime (Finlayson-Pitts, 2003). However, as shown in Fig. S6, the contribution of HCl to Cl radicals is much lower than the photolysis of ClNO and ClNO₂. Such small contribution of HCl were also reported in a box-model study in North China Plain (Liu et al., 2017). It suggests that the limited role of these thermodynamic equilibrium reactions in the Cl radicals and following O₃ formation."

**Reviewer response: Thank you for addressing this comment.**

[Comment 6] Line 262: You mention increases in Cl radicals after sunrise are more pronounced at higher altitudes. Should there be something to direct readers to Fig. 5? I did not see Fig. 5 mentioned in the text, but it is possible I missed it. Can you be more clear by what you mean that 'increases in Cl radicals are more pronounced at higher altitudes shortly after sunrise'? In Fig. 5, I see the changes in Cl concentrations after sunrise are pretty similar in the boundary layer compared to right above the boundary layer for the three cities. Changes in Cl radical concentrations above 2 km are often 0.

Response: Thanks for your carefully review. This is a typo and we mistakenly removed some words of this paragraph in the original manuscript. In the revised manuscript, we have mentioned Fig. 5 in the text to explain the diurnal and vertical changes in Cl radicals induced by SSA. Besides, we have provided a clearer description and discussion in the vertical changes of Cl radicals.

Revision in the updated manuscript:

(1) Line 277: "Furthermore, the impact of SSA on Cl radicals is observed not only at the surface but

also vertically through the atmosphere. Figure 5 examines the vertical-diurnal variations in SSAinduced Cl radical concentrations in Beijing, Shanghai, and Guangzhou."
(2) Line 280: "These increases are more pronounced near the top of planetary boundary layer shortly after sunrise, suggesting the impact of SSA on Cl radicals is more significant in upper levels than near the surface."

**Reviewer response: Thank you for addressing this comment. It is clear now what you mean.**

[Comment 7] Lines 275 – 286: It's interesting that J(NO$_2$) was decreased considerably in the upper troposphere. That high, you would presumably have higher actinic flux and less scattering from sea salt than at the surface. Fig 1 shows that changes in sea salt particle mass (using Na+ as a proxy) in the upper troposphere are close to zero. Are these low mass concentrations enough to scatter enough radiation to reduce J(NO$_2$) to the same degree as it is reduced at the surface, especially considering the higher actinic flux aloft? Perhaps some discussion here to elaborate on the result would be useful and of interest.
Response: Thanks for your valuable comment. The effect of aerosol particles on the extinction depends on their size distribution. Generally, fine particles have higher extinction effect than coarse particles (Molnár and Mészáros, 2001), and it can be transported to the higher levels. As a result, despite low concentrations in upper levels (2-3 km), fine SSA can reduce J(NO$_2$) to the same degree as those near the surface. We have provided some discussions in the revised manuscript.
Revision in the manuscript:
(1) Line 303: "We note that the extinction effect of SSA can extend into the upper levels (2-3 km), where the decrease in J(NO$_2$) can be the same degree as those observed near the surface. This is because the aerosol extinction effect depends on particle size distribution. Fine SSA particles have higher extinction effect than coarse ones (Molnár and Mészáros, 2001), and it can be transported to the higher levels."

**Reviewer response: Thank you for responding to this comment and for the Molnar and Meszaros citation. I looked at this paper, and it does not specifically mention that fine SSA particles have a higher extinction effect compared to coarse SSA particles. The work you are referencing sampled particles from 10 m on the Great Hungarian Plain, which appears to be inland and far from marine sources. To me it seems unlikely they sampled considerable mass concentrations of sea salt particles, and when searching for the words 'sea salt' and 'marine' in the paper, nothing comes up. Thus, I recommend you revise your citation of Molnar and Meszaros (2001) when stating fine SSA have a higher extinction effect than coarse SSA. Perhaps you can find another work explicitly mentioning SSA in this context. Alternatively, Molnar and Meszaros (2001) do support your statement that fine particles are more dominant in governing extinction, so perhaps you can write this without explicitly mentioning SSA in the statement.**

[Comment 8] Lines 291 – 294: Correct me if I'm wrong but Fig. 7 shows differences between the NOSA and BASE simulations. Thus, wouldn't the VOC concentrations be the same over the remote oceanic regions in both scenarios? If not, please explain why. If not, then perhaps the main explanation for lower HO$_2$ concentrations over remote oceanic regions is reduced photolysis due to scattering by sea salt particles? It's impressive the reduction in photolysis due to light scattering is enough to decrease production of HO$_2$ when Cl-radicals from chloride depletion reactions are considered in the model. Fig. 4 shows that there were considerable increases in Cl- radicals over these remote oceanic regions, yet the competing effect of scattering by sea salt particles seems to have countered any increases in HO$_2$ that would be caused by additionally available Cl- radicals. As mentioned earlier, it would be great to understand and/or mention the robustness and accuracy of the changes in photolysis rates due to increased SSA scattering in the model since this is such a prominent topic/result of the paper.
Response: Thank you for your valuable suggestion. The VOC concentrations are the same over the remote oceanic regions in both scenarios. We agree that the decreased HO$_2$ over remote oceanic

regions is mainly attributed to the reduced photolysis rates caused by SSA. Please see our responses to the 1st comment for the discussions of the CMAQ module to calculate the photolysis rates.
Revision in the manuscript:
(1) Line 319: "In remote oceanic regions, where VOC concentration is generally low, a decrease in $HO_2$ can be observed. This decrease is mainly attributed to the reduced photolysis rates due to the extinction effect of SSA, which seems to have countered any increases in $HO_2$ that would be caused by additionally available Cl radicals (Fig. 4c).

**Reviewer response: Thank you for addressing this comment.**

[Comment 9] Lines 309-312: Similar comment as above. The decreases in OH concentrations in the troposphere are interesting considering that sea salt particle mass concentrations are presumably low at those altitudes and actinic fluxes are stronger. I know the transport of SSA is possible above the boundary layer as you mentioned, but changes in SSA above 3 km are mostly zero across all months. It is interesting that the scattering from such relatively low amounts of SSA is enough to offset any potential increases in OH due to increased presence of Cl- radicals (although the change in Cl- radicals above 3 km is also close to 0 for all months). There is not really an action item for this comment, I just found it interesting and I think the results would be strengthened by mentioning/citing the validity of the simulated changes in photolysis rates due to increased SSA scattering at least once somewhere in the paper.
Response: Thanks for your valuable suggestion. There are some previous studies verified and evaluated the module to calculate the photolysis rates in CMAQ. We have cited these references in the manuscript. Please also see our responses to the above 1st and 7th comment.

**Reviewer response: Thank you for addressing this comment.**

[Comment 10] Lines 335 – 356: The relationship between the sign of the change in $O_3$ and whether or not the region is NOx- or VOC-limited is interesting. Previously, reductions in concentrations of various radicals and species involved with the NOx-VOC-O3 system of reactions were attributed primarily to changes in photolysis rates due to scattering by sea salt particles. I wonder now if whether those altitudes are NOx- or VOC-limited may be of interest to consider for explaining reductions in the concentrations of those species at higher altitudes? The regime is mentioned when considering the vertical profile in changes of $O_3$ concentrations, so I wonder if a similar discussion of regime would be appropriate to at least mention for other species involved in reactions related to the production of $O_3$?
Response: Thank you for your valuable suggestion to further explore the role of $NO_x$/VOC-limited regimes in explaining altitude-dependent changes in ozone-related species. Higher levels are generally in $NO_x$-limited conditions due to lower $NO_x$ concentrations (Wang et al., 2025; Lin et al., 2022), and the SSA-induced decrease in $NO_x$ can reduce $O_3$ formation. We have added some discussions in the revised manuscript.
Revision in the manuscript:
(1) Line 386: "We also find significant decreases in SSA-induced $O_3$ concentrations over oceanic regions (Fig. 10) and in the upper levels (Fig. 11). This decline can be explained by two reasons: For one thing, remote oceanic areas (Fig. 10c) and upper levels (Wang et al., 2025; Lin et al., 2022) are generally in $NO_x$-limited conditions due to lower $NO_x$ concentrations, and the SSA-induced decrease in $NO_x$ (Fig. 3) reduce $O_3$ formation; for another, in these areas with scant VOCs, SSAinduced Cl radicals preferentially react with $O_3$ to form ClO (as depicted in Fig. S9 and S10), which enhances $O_3$ depletions. This behavior mirrors stratospheric conditions where Cl radicals are pivotal in consuming $O_3$."

**Reviewer response: Thank you for addressing this comment.**

[Comment 11] Line 260: Typo? Says "Furthe es"

Response: Thanks for pointing out this typo. We mistakenly removed some words of this paragraph in the original manuscript. We have corrected this issue.
Revision in the updated manuscript:
(1) Line 277: "Furthermore, the impact of SSA on Cl radicals is observed not only at the surface but also vertically through the atmosphere. Figure 5 examines the vertical-diurnal variations in SSAinduced Cl radical concentrations in Beijing, Shanghai, and Guangzhou."

**Reviewer response: Thank you for addressing this comment.**

**Additional reviewer comments:**

**On line 153 you now state**

'This parameterization was identical across different aerosol modes (Aitken, accumulation, and coarse).'

**and on line 192, you now mention the composition of dry SSA in different modes remains consistent. Would you mind adding the exact size range of SSA considered in your model (e.g., XX – XX nm)?**

**There are several instances of issues with subject-verb agreement still in the manuscript. Hopefully additional editing stages will handle these issues.**

---

## Author Response (AR2)

Response to the comment from Anonymous referee #1

[1] Reviewer response: Thank you for addressing this. First, do you mean "chloride depletion of SSA through…" as opposed to 'chloride deposition of SSA through…"

I see that your model results show the production of Cl radicals from ClNO and $ClNO_2$ to be more important than production from reactions involving $H_2SO_4$, $HNO_3$, and SSA. If I understand correctly, reactions forming ClNO and $ClNO_2$ would result in a replacement of particulate $Cl^-$ with $NO_3^-$ in sea salt particles. Thus if pathways involving ClNO and $ClNO_2$ are the dominant way in which reactive Cl in SSA is being released to the atmosphere, would you expect the composition of depleted SSA to contain mostly $NO_3^-$ with relatively small amounts of $SO_4^{2-}$? I ask because studies looking at the composition of aged SSA such as Braun et al. (2017; https://doi.org/10.1021/acs.est.7b02039) and AzadiAghdam et al. (2019; https://doi.org/10.1016/j.atmosenv.2019.116922) show that a considerable amount of chloride depletion in submicron SSA can be attributed to $SO_4^{2-}$ in certain conditions. Braun et al. is considering a different region, while AzadiAghdam et al. features data from the Philippines. I have seen other studies as well attributing non-negligible amounts of chloride depletion to reactions between $H_2SO_4$ and SSA when analyzing the composition of aged sea salt particles. Your study is looking at it from a different perspective (where you are NOT studying chloride depletion from the perspective of the chemical composition of the sea salt particles themselves), so I wonder how to resolve the differences between the perspectives when it comes to the role played by $HNO_3$ and $H_2SO_4$. I certainly am not casting doubt on your model results, and I see they show changes in $SO_2$ are small and that HCl is not a dominant contributor to Cl radicals.

I think researchers studying chloride depletion from the perspective of SSA chemical composition will find your results interesting regarding the chemical pathways leading to the presence of $HNO_3$ in aged sea salt particles. I think currently many researchers see $HNO_3$ in aged SSA and report that the $HNO_3$ came to exist in the SSA through the pathway $HNO_3 + NaCl \rightarrow HCl + NaNO_3$, whereas you show that other pathways involving $N_2O_5$ and $NO_2$ may lead more dominantly to the presence of $NaNO_3$ in SSA (if I understand your work correctly, please correct me if I'm misunderstanding). For example, here is a quote from Su et al. (2022; https://doi.org/10.1016/j.atmosenv.2022.119365):

"The acid displacement reaction is regarded as a major chemical pathway in chloride depletion,

releasing HCl:HA (g or aq) + NaCl (aq or s) → NaA (aq or s) + HCl (g or aq)where HA (g or aq) represents acidic species (e.g., sulfuric acid,nitric acid, organic acid) in gaseous (g) or aqueous (aq) phases. NaCl (aq or s) is the major component of SSA in aqueous or solid (s) phases. NaA is the sodium salt from the interaction between HA and NaCl. It should be noted that the acid displacement reaction involving organic acid in SSA is reversible."

I apologize for the long-winded comment. I just find it interesting that from the modeling perspective Cl radical formation from HCl is small compared to that from ClNO and $ClNO_2$, whereas in sea salt chemistry papers, the main reaction presented in describing the process of Cl-depletion is HA + NaCl → NA + HCl.

This paper showcases how it is important to consider (1) modeling studies looking at the most important pathways regarding the production of Cl radicals and (2) studies examining chloride depletion from the perspective of SSA chemical composition as the two perspectives may be inclined to emphasize different reaction pathways in explaining the presence of $HNO_3$ in aged SSA. No action needed for this comment, just wanted to put these thoughts out there and see if you had any thoughts on this. I am more familiar with the particle chemistry side, so it would be interesting to hear your perspective from the modeling side.

Response: Firstly, thanks for pointing out this typo. We have replaced "chloride deposition" with "chloride depletion" in the manuscript.

Secondly, we sincerely appreciate this insightful comment, which illuminates a fascinating methodological duality in interpreting sea salt chloride depletion mechanisms. Below, we integrate your observations with our model results to reconcile these viewpoints:

(1) The relative importance of sulfate vs. nitrate in chloride depletion can be regionally heterogeneous. In high-$SO_2$ regions, sulfate accumulation via $H_2SO_4$ + 2NaCl → $Na_2SO_4$ + 2HCl may dominate chloride loss. In high-$NO_x$ regions, nitrate pathways ($N_2O_5$ hydrolysis: $N_2O_5$ + NaCl → $NaNO_3$ + $ClNO_2$; or $HNO_3$ + NaCl → $NaNO_3$ + HCl) prevail, explaining the prominence of particulate $NO_3^-$ over $SO_4^{2-}$. This spatial variability reconciles the seemingly contradictory literature – both pathways operate, but their relative weights depend on local precursor emissions.

(2) The small contribution of HCl to Cl radicals in our model may not imply low HCl production from acid displacement reactions (HA + NaCl → NaA + HCl). Rather, it reflects that the HCl + OH → Cl + $H_2O$ reaction is slow, especially compared to rapid $ClNO/ClNO_2$ photolysis. Thus, even

sufficient HCl production may not yield significant Cl radicals.

While quantifying sulfate/nitrate partitioning in depleted SSA was not this study's primary focus, we agree this represents a compelling research frontier. We will conduct targeted sensitivity experiments in future work (e.g., zero-out $H_2SO_4$/$HNO_3$ emissions) to isolate their individual impacts on chloride depletion products.

Revision in the manuscript:

(1) Line 144: "The chlorine depletion of SSA through its equilibrium reactions with $H_2SO_4$ and $HNO_3$ were considered in the model"

[2] Reviewer response: Thank you for responding to this comment and for the Molnar and Meszaros citation. I looked at this paper, and it does not specifically mention that fine SSA particles have a higher extinction effect compared to coarse SSA particles. The work you are referencing sampled particles from 10 m on the Great Hungarian Plain, which appears to be inland and far from marine sources. To me it seems unlikely they sampled considerable mass concentrations of sea salt particles, and when searching for the words 'sea salt' and 'marine' in the paper, nothing comes up. Thus, I recommend you revise your citation of Molnar and Meszaros (2001) when stating fine SSA have a higher extinction effect than coarse SSA. Perhaps you can find another work explicitly mentioning SSA in this context. Alternatively, Molnar and Meszaros (2001) do support your statement that fine particles are more dominant in governing extinction, so perhaps you can write this without explicitly mentioning SSA in the statement.

Response: Thanks for your valuable comment. We agreed that Molnár and Mészáros (2001) did support the statement that fine particles are more dominant in governing extinction. To avoid misleading, in the revised manuscript, we stated this without explicitly mentioning SSA.

Revision in the manuscript:

(1) Line 307: "Fine particles have a higher extinction effect than coarse ones (Molnár and Mészáros, 2001), and they can be transported to higher levels"

Additional reviewer comments:

[3] On line 153 you now state 'This parameterization was identical across different aerosol modes (Aitken, accumulation, and coarse).' and on line 192, you now mention the composition of dry SSA in different modes remains consistent. Would you mind adding the exact size range of SSA considered in your model (e.g., XX – XX nm)?

Response: Thanks for your valuable comment. According to the codes in the CMAQ model, the exact size range of SSA is from ~0.02 to 20 μm. We have added this information in the manuscript.

Revision in the manuscript:

(1) Line 180: "The estimated diameters of SSA range from ~0.02 μm to 20 μm in the model."

[4] There are several instances of issues with subject-verb agreement still in the manuscript. Hopefully additional editing stages will handle these issues.

Response: We sincerely appreciate the reviewer's careful attention to grammatical details. All instances of subject-verb agreement have been thoroughly reviewed and corrected in the revised manuscript.